# LLaVA-KD: A Framework of Distilling Multimodal Large Language Models

## Abstract

The success of Large Language Models (LLM) has led researchers to explore Multimodal Large Language Models (MLLM) for unified visual and linguistic understanding. However, the increasing model size and computational complexity of MLLM limit their use in resource-constrained environments. Small-scale MLLM ($s$-MLLM) aims to retain the capabilities of the large-scale model ($l$-MLLM) while reducing computational demands, but resulting in a significant decline in performance. To address the aforementioned issues, we propose a novel LLaVA-KD framework to transfer knowledge from $l$-MLLM to $s$-MLLM. Specifically, we introduce Multimodal Distillation (MDist) to minimize the divergence between the visual-textual output distributions of $l$-MLLM and $s$-MLLM, and Relation Distillation (RDist) to transfer $l$-MLLM's ability to model correlations between visual features. Additionally, we propose a three-stage training scheme to fully exploit the potential of $s$-MLLM: *1)* Distilled Pre-Training to align visual-textual representations, *2)* Supervised Fine-Tuning to equip the model with multimodal understanding, and *3)* Distilled Fine-Tuning to further transfer $l$-MLLM capabilities. Our approach significantly improves performance without altering the small model's architecture. Extensive experiments and ablation studies validate the effectiveness of each proposed component. Code will be available.

## 1 Introduction

Inspired by the significant achievements of Large Language Models (LLM) in the field of Natural Language Processing, an emerging and rapidly developing research area is focusing on the development of Multimodal Large Language Models (MLLM). These models integrate visual encoder, feature projector, and LLM to achieve a unified understanding of visual and linguistic information. However, the success of LLMs benefits from the scaling law, which significantly increases the model size. The large-scale model and high-cost inference limit the application of MLLMs in resource-constrained scenarios. To solve this challenging problem, some studies (Zhu et al., 2024; Chu et al., 2023) have attempted to reduce model scale by directly adopting lightweight LLMs, but this reduction often comes with a significant decline in model performance. Some methods compensate for this issue by optimizing model structure and improving the quality of training data, *e.g.,* MoE-LLaVA (Lin et al., 2024) introduces the Mixture-of-Experts (Jacobs et al., 1991) (MoE) to enhance the model's ability for complex multimodal information while maintaining the computational cost of the lightweight LLM, and Bunny (He et al., 2024) improves the training data quality by removing redundant data. Unlike these methods, we explore improving the performance of the small-scale MLLM ($s$-MLLM) from the perspective of investigating various training strategies without altering the model architecture. As shown in Fig. 1(a), current $s$-MLLM follow the two-stage training strategy of the large-scale MLLM ($l$-MLLM), which includes Pre-Training (PT) and Supervised Fine-Tuning (SFT). The PT stage is used to project visual features to the text embedding space, while the SFT stage is used to enhance the model's understanding and reasoning capabilities. However, due to the limited model capacity, using the same training strategy as $l$-MLLM may prevent $s$-MLLM from effectively learning the complex knowledge that $l$-MLLM can capture (Kaplan et al., 2020). Knowledge distillation, as a model compression technique, has proven its effectiveness in traditional visual tasks. However, the application of knowledge distillation to MLLM has not been fully explored. In this paper, we investigate how knowledge distillation can be leveraged to improve the training of $s$-MLLM.

Figure 1: To train a small-scale MLLM, (a) the existing methods follow a two-stage training scheme, including Pre-Training (PT) and Supervised Fine-Tuning (SFT). (b) Our LLaVA-KD proposes a three-stage scheme to exploit the potential of $s$-MLLM, including Distilled Pre-Training (DPT) to align visual-textual representation, SFT to equip the model with multimodal understanding, and Distilled Fine-Tuning (DFT) to transfer $l$-MLLM's capacities. (c) This study compares our LLaVA-KD with several SoTA MLLMs on five popular multimodal benchmarks.

Essentially, MLLM leverages LLM for multimodal information understanding and reasoning. Therefore, the core of distillation in MLLM involves transferring multimodal information from the $l$-MLLM to the $s$-MLLM based on LLM. Previous research on LLM distillation (Gu et al., 2024; Ko et al., 2024) primarily employs the standard Kullback-Leibler Divergence (KLD) to minimize the discrepancy in output distributions of responses between the $l$-MLLM and $s$-MLLM, thereby promoting the $s$-MLLM to obtain more accurate responses. However, in the context of MLLM, effective visual representations can promote the multimodal information understanding, thereby further improving the quality of responses. Therefore, we extend the distillation process to include the visual distribution, using KLD to minimize discrepancies in both visual and language modalities. Furthermore, to enhance the $s$-MLLM's ability to model the contextual relationships of visual representations, we introduce Relation Distillation (RDist). This technique transfers the $l$-MLLM's ability to model the correlations between visual representations to the $s$-MLLM. By distilling multimodal information from both visual and language modalities (MDist) and incorporating RDist, we can achieve a more comprehensive and effective multimodal knowledge transfer.

In the common PT-SFT two-stage training scheme, MLLM primary acquires the understanding capacity through the SFT stage. Therefore, a straightforward approach is to introduce knowledge distillation during the SFT stage, to enhance $s$-MLLM's capacities. However, we find this scheme to be suboptimal. In this paper, we propose an improved three-stage training strategy, as shown in Fig. 1(b). Firstly, in MLLM, aligning the visual representation with textual representation is a prerequisite for multimodal information understanding. To promote this goal, we propose a novel approach, incorporating the distillation during the PT stage, utilizing $l$-MLLM to guide the predictions of $s$-MLLM. In this way, $s$-MLLM not only improves the accuracy of predictions but also further optimizes the alignment between visual and language modalities. Secondly, we observe that applying knowledge distillation at the SFT stage is insufficient for the $s$-MLLM to fully acquire the capabilities of the $l$-MLLM. To address this, we introduce a "SFT-DFT" shceme. Specifically, we first initilize the $s$-MLLM with understanding and reasoning capabilities through SFT. Subsequently, we use DFT to achieve the transfer of capabilities from $l$-MLLM to $s$-MLLM.

Compared to the current advancements in s-MLLM, our method exhibits impressive performance in various multimodal benchmarks. For instance, as illustrated in Fig. 1(c), LLaVA-KD-2B comprehensively outperforms recent s-MLLMs such as Imp (Shao et al., 2024), Bunny (He et al., 2024), and TinyLLava (Zhou et al., 2024). We summarize our contributions as follows:

- We introduce LLaVA-KD, a novel MLLM-oriented distillation framework to transfers the knowledge from large-scale MLLM to the small-scale MLLM. Specifically, it contains a three-stage distillation scheme, including Distilled Pre-Training (DPT) to enhance the multimodal alignment process, as well as Supervised Fine-Tuning (SFT) and Distilled Fine-Tuning (DFT) to effectively transfer capacities from the large to small MLLM.

- We propose an innovative distillation strategy that combines Multimodal Distillation (MDist) with Relational Distillation (RDist). Both them are used in the DPT and DFT stages to enhance the ability of $s$-MLLMs to process complex visual information.

- We demonstrate the superiority and efficiency of LLaVA-KD. Our model significantly surpasses the recent small-scale MLLM advancements such as Imp and Bunny on nine popular multimodal benchmarks.

## 2 RELATED WORKS

### 2.1 MULTIMODAL LARGE LANGUAGE MODEL

With the development of LLM, researchers have turned their attention to MLLM to promote the understanding of vision-language cross-modal information. BLIP-2 (Li et al., 2023a) trains a Querying Transformer through various image-text tasks to bridge the modality gap. Flamingo (Alayrac et al., 2022) integrates visual features into LLM through gated attention. Recent methods (Liu et al., 2024b;a; Bai et al., 2023) align visual features with textual features through a projector such as Multi-Layer Perceptron (MLP) or Q-Former (Li et al., 2023a). Then they will enhance the model's instruction-following ability through supervised instruction-tuning, making MLLMs better meet human needs. One research trend is to further enhance the fine-grained visual perception ability of MLLM by enabling the model to support high-resolution inputs (Li et al., 2024; Luo et al., 2024), so that MLLMs can be widely applied to various downstream tasks such as image segmentation and grounding. Although the aforementioned methods have shown great potential in visual understanding tasks, their large model size and computational cost greatly limit the application of the model in resource-constrained scenarios, such as mobile devices.

**Lightweight Multimodal Large Language Model**  Existing lightweight MLLMs mainly reduce model parameters by employing lightweight LLMs. For example, LLava-Phi (Zhu et al., 2024) follows the model structure of LLaVA1.5 (Liu et al., 2024a) and replaces LLMs with the lightweight Phi-2; Some work has shown that optimizing model structure and training data can compensate for performance degradation caused by reduced model capacity. MoE-LLaVA (Lin et al., 2024) introduces MoE into LLMs, showing potential in multimodal understanding and hallucination suppression with only 3B activation parameters. Bunny (He et al., 2024) performs K-Means clustering on the image embeddings derived from the LAION-2B dataset. Subsequently, it constructs an undirected graph to filter out images with excessively high similarity. This process not only enriches the information but also effectively reduces the size of the training set. Unlike these methods, our approach primarily focuses on improving the training strategy of MLLMs. In this paper, we propose a three-stage training recipe based on knowledge distillation. By transferring the knowledge of large MLLMs to lightweight MLLMs, the Light MLLMs' capabilities will be significantly enhanced.

### 2.2 KNOWLEDGE DISTILLATION

Knowledge Distillation (KD) (Hinton, 2015) aims to transfer the knowledge from a large, complex teacher model to a lightweight, simple student model. This technique can significantly improve the performance of small models with fewer parameters, less computation, and faster speed. Knowledge distillation has been successful applied in visual tasks and has achieved success in many fields, typically in the domain of image classification. For example, traditional distillation methods (Hinton, 2015) use soft logits of the teacher model as extra supervision to train the student model. DKMF (Wang et al., 2021) and FNKD (Xu et al., 2020) reveal that mimicking the teacher model's features leads to more accurate classification. DGKD (Son et al., 2021) further improves the student model's predictions by integrating multiple teacher models for guidance.

**KD for LLM.**  With the successful release of ChatGPT and its significant application value, LLM has gradually attracted attention and achieved numerous research progress in recent years (Brown, 2020; Achiam et al., 2023). However, to achieve better results, the model size has also become increasingly larger which follows scaling law (Kaplan et al., 2020), which limits the application of LLM in resource-constrained scenarios. Therefore, some researchers have recently begun to explore the application of knowledge distillation in LLM.

MiniLLM (Gu et al., 2024) and DistiLLM (Ko et al., 2024) are dedicated to optimizing distillation process, proposing reverse Kullback-Leibler Divergence (KLD) and skew KLD respectively, to prevent the student model from overly focusing on the long-tail distribution of the teacher model's output. (Wu et al., 2024) proposes a strategy to adaptively balance the weights of KLD and reverse KLD loss. Some methods (Hsieh et al., 2023; Tian et al., 2024; Ranaldi & Freitas, 2024) use the Chain-of-Thought (CoT) capability of large LLMs to model causal relationships, and enrich training data. Considering that different LLMs have different reasoning capabilities, TinyLLM (Tian et al., 2024) used multiple teacher models during training.

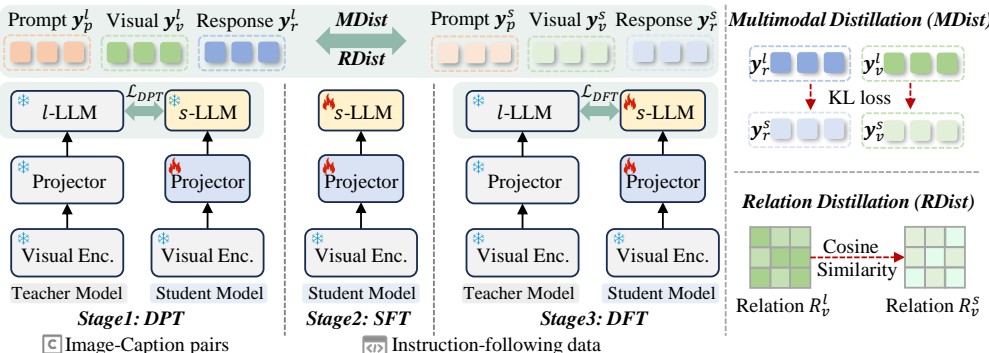

Figure 2: **Overview of our LLaVA-KD** that contains three stages for effect training: *1)* Distilled Pre-Training (DPT) to align visual and text information as $l$-MLLM. *2)* Supervised Fine-Tuning (SFT) to enable $s$-MLLM with multimodal understanding capacity. *3)* Distilled Fine-Tuning (DFT) to transfer $l$-MLLM's capacities to $s$-MLLM. During the training phase, we employ Multimodal Distillation (MDist) in both DPT and DFT stages, and develop Relation Distillation (RDist) to enable $s$-MLLM to capture the complex relationships in visual information.

**KD for MLLM.** Most recently, LLAVA-MoD (Shu et al., 2024) applies knowledge distillation to train $s$-MLLM. It first optimizes the structure of $s$-MLLM by integrating MoE (Jacobs et al., 1991; Lin et al., 2024) into the LLM, thereby enhancing the model's expressive ability. For model training, it firstly uses standard KLD to align the output response logits distribution between the $s$-MLLM and $l$-MLLM. Additionally, it introduces a preference distillation process to improve the $s$-MLLM's judgment capability, thereby reducing hallucinations. LLaVADI (Xu et al., 2024) is another $s$-MLLM work based on distillation, which reveals that most training strategies designed for LLMs do not bring additional benefits to the MLLMs. Meanwhile, they propose that using teacher models for data augmentation is beneficial to promote the learning of student models.

Unlike existing LLM/MLLM distillation methods, which design complex constraints, introduce multi-teacher models to enhance supervision, or explore complicated model structures, we focus on optimizing training schemes and developing multimodal distillation strategies, to effectively and efficiently improve the performance of existing small-scale MLLM under a single-teacher model.

## 3 LLaVA-KD

The deployment of lightweight MLLMs is crucial for resource-constrained environments. However, small-scale MLLMs trained using naive strategies often yield suboptimal results. For instance, a 4B model of TinyLLaVA achieves 65.0%, while reducing the LLM to 0.5B only results in 54.7%, which exhibits a significant performance gap. To address this issue, we propose an innovative three-stage training scheme with the novel distillation strategy termed LLaVA-KD in Fig. 2.

### 3.1 COMPOSITION OF DISTILLED MLLM ARCHITECTURE

Fig. 2(Left) illustrates the distillation process for MLLM, which includes a large-scale $l$-MLLM as the teacher model and a small-scale $s$-MLLM as the student model. Both them follow the simple design of LLaVA-1.5 (Liu et al., 2024a), and each includes three main components:
**Frozen Visual Encoder** is used to obtain powerful visual features, and we employ the pre-trained SigLIP (Zhai et al., 2023) following previous success (He et al., 2024; Tong et al., 2024). Specifically, the given input image $X_v \in \mathbb{R}^{H \times W \times 3}$ is first sequenced to 2D patches $P_v \in \mathbb{R}^{N_p \times S_p^2 \times 3}$ with $S_p$ and $N_p$ representing patch size and its number, respectively. The final transformer layer projects $P_v$ to visual features $Z_v \in \mathbb{R}^{N_p \times C}$ that the feature dimension is $C$. Both teacher and student models use the same visual encoder by default.

**Visual Projector** contains two MLP layers with a GELU activation function to project visual features $Z_v$ into the text embedding space $H_v \in \mathbb{R}^{N_p \times D}$, where $D$ denotes the embedding dimensions.

**Large Language Model (LLM)** is used to achieve unified understanding of visual and linguistic information. Given the multimodal input of visual embedding $H_v$ and text embedding $H_t$, the LLM takes their concatenation $H = [H_v, H_t]$ as input to generate the output $\mathbf{y} = [\mathbf{y}_p, \mathbf{y}_v, \mathbf{y}_r] = \{y_t\}_{t=1}^{T}$, where $\mathbf{y}_p$, $\mathbf{y}_v$, and $\mathbf{y}_r$ denote prompt, visual, and response tokens, and $T$ denotes the length of all prediction tokens. Specifically, we denote teacher and student LLMs as $l$-LLM and $s$-LLM.

## 3.2 TRAINING SCHEME OF TEACHER MODEL $l$-MLLM

We introduce the common training scheme for powerful $l$-MLLMs, which is regarded as the performance upper limit of $s$-MLLM. This scheme consists of two stages, as described in TinyLLaVA (Zhou et al., 2024):

**Pre-Training.** The *Visual Encoder* and $l$-*LLM* are kept frozen, and only the *Projector* is optimized to align visual features with textual features. During training, we use image-caption pairs and corresponding objective is formulated as:

$$\mathcal{L}_{reg} = - \sum_{m=1}^{M} \log \phi_l \left( y_m \mid \mathbf{y}_{<m} \right), \tag{1}$$

where $M$ denotes the length of predicted response tokens, while $\phi_l \left( y_m \mid \mathbf{y}_{<m} \right)$ represents the distribution of the response token $y_m$ based on the condition of previous predictions $\mathbf{y}_{<m}$.

**Supervised Fine-Tuning.** This stage keeps the *Visual Encoder* frozen, aiming at jointly optimizing *Projector* and $l$-*LLM* to enhance understanding and instruction-following capacities of the teacher model $l$-MLLM. During training, we leverage high-quality conversation datasets and the training objective $\mathcal{L}_{SFT}$ is described in Eq. 1.

## 3.3 FRAMEWORK OF LLAVA-KD

For the large-scale teacher model, we adopt the previous training strategy (Sec. 3.2) to develop the $l$-MLLM. For training $s$-MLLM, we propose a novel distillation strategy tailored for multimodal information learning (Sec. 3.3.1), and we further design a three-stage distillation scheme (Sec. 3.3.2).

### 3.3.1 MLLM-ORIENTED KD STRATEGY

**Multimodal Distillation (MDist).** Considering that MLLM essentially leverages LLM for multimodal information understanding and reasoning, we follow the naive distillation method of LLM () that uses Kullback-Leibler Divergence (KLD) to distill the response predictions. The training objective can be defined as:

$$\begin{aligned}
\mathcal{L}_{res} &= \sum_{m=1}^{M} \mathrm{KLD}(\phi_l(y_m \mid \mathbf{y}_{<m}), \phi_s(y_m \mid \mathbf{y}_{<m})), \\
&= \sum_{m=1}^{M} \sum_{j=1}^{V} \phi_l \left( Y_j \mid \mathbf{y}_{<m} \right) \log \left( \frac{\phi_l \left( Y_j \mid \mathbf{y}_{<m} \right)}{\phi_s \left( Y_j \mid \mathbf{y}_{<m} \right)} \right),
\end{aligned} \tag{2}$$

where $M$ represents the length of response tokens and $V$ denote and vocabulary space. $\phi_l$ and $\phi_s$ denote the model parameters of $l$-MLLM and $s$-MLLM, respectively, $\phi_l \left( Y_j \mid \mathbf{y}_{<m} \right)$ and $\phi_s \left( Y_j \mid \mathbf{y}_{<m} \right)$ denote the probability of vocabulary $Y_j$ in the response token $y_m$, as predicted by $l$-MLLM and $s$-MLLM.

Meanwhile, the visual representation is also critical for multimodal understanding of LLM. Therefore, we further optimize the KLD between the output visual distribution of the teacher and student:

$$\mathcal{L}_{vis} = \sum_{k=1}^{K} \sum_{j=1}^{V} \phi_l \left( Y_j \mid \mathbf{y}_{<k} \right) \log \left( \frac{\phi_l \left( Y_j \mid \mathbf{y}_{<k} \right)}{\phi_s \left( Y_j \mid \mathbf{y}_{<k} \right)} \right), \tag{3}$$

where $K$ denotes the length of visual tokens, $\phi_l \left( Y_j \mid \mathbf{y}_{<k} \right)$ and $\phi_s \left( Y_j \mid \mathbf{y}_{<k} \right)$ denote the probability of vocabulary $Y_j$ in the visual token $y_k$, as predicted by $l$-MLLM and $s$-MLLM.

We utilize MDist in the DPT stage to facilitate the alignment of visual and language features in $s$-MLLM, while enhancing the $s$-MLLM's understanding and reasoning capabilities in the DFT stage.

**Relation Distillation (RDist).** To enable the student model to capture the complex relationships in visual information, we construct a self-correlation matrix from the visual tokens output by the LLM. By optimizing the similarity between matrices, the student model inherits the teacher model's ability to comprehend the intricate relationships among visual tokens. To achieve this, we first compute the self-correlation matrices as follows:

$$\begin{cases} R_v^s = \mathbf{y}_v^s \otimes \mathbf{y}_v^s \in \mathbb{R}^{N_p \times N_p}, \\ R_v^t = \mathbf{y}_v^t \otimes \mathbf{y}_v^t \in \mathbb{R}^{N_p \times N_p}, \end{cases} \tag{4}$$

where $\otimes$ represents matrix multiplication, $\mathbf{y}_v^s$ and $\mathbf{y}_v^t$ denote the visual logits of the student and teacher, and $N_p$ denotes the number of visual tokens. Following this, we maximum the cosine similarity between the $R_v^s$ and $R_v^t$ that is formulated as:

$$\mathcal{L}_{rel} = 1 - \mathrm{Cos}(R_v^s, R_v^t) = 1 - \frac{R_v^s \cdot R_v^t}{\|R_v^s\|\|R_v^t\|}, \tag{5}$$

where $\mathrm{Cos}(\cdot)$ denotes the cosine similarity. We use RDist to further improve visual representations in $s$-MLLM at both DPT and DFT stages.

### 3.3.2 THREE-STAGE DISTILLATION SCHEME

Based on the existing training scheme for MLLMs, a straightforward idea is that introducing knowledge distillation during the SFT stage can effectively enhance model performance. However, our research indicates that this training scheme is suboptimal (Refer to Table 2). Therefore, we consider whether it is feasible to introduce a distillation strategy during the pre-training phase or to design additional fine-tuning distillation to improve the performance of $s$-MLLM, and finally we propose a novel and powerful three-stage training scheme.

**Distilled Pre-Training (DPT).** The main purpose of this stage is to project visual features to the text embedding space. Previous methods (Liu et al., 2024a; Zhu et al., 2024) primarily achieve this by optimizing the autoregressive prediction process of LLM (Eq. 1). In our LLaVA-KD, we utilize a distillation procedure to better align visual and textual information as $l$-MLLM.

Specifically, we freeze the visual encoder and LLM of $s$-MLLM, and only optimize the projector. During the training process, we minimize the discrepancy between the student model and the teacher model in terms of the output distribution of visual and response through MDist. To optimize this objective, the alignment of the projected visual features with the text embedding can be further promoted. Furthermore, we utilize RDist to enhance the quality of visual features by enabling the student model to learn from the teacher model's ability to handle complex visual information.

Overall, in addition to optimizing the autoregressive prediction results, we also utilize a multimodal distillation and relation distillation procedure. The objective is defined as follows:

$$\mathcal{L}_{DPT} = \textcolor{red}{\mathcal{L}_{reg}} + \alpha\mathcal{L}_{res} + \beta\mathcal{L}_{vis} + \gamma\mathcal{L}_{rel}, \tag{6}$$

where $\alpha$, $\beta$, and $\gamma$ are weights of each objective item.

**Supervised Fine-Tuning (SFT).** At this stage, we follow the common SFT procedure of the large MLLM's training phase (Sec. 3.2). By jointly training the Projector and $l$-LLM, we initialize the model with reasoning ability and instruction-following capability. The training objective can be defined as Eq. 1, denoted as $\mathcal{L}'_{SFT}$.

**Distilled Fine-Tuning (DFT).** The main objective of this stage is to further enhance the understanding and reasoning capacities of $s$-MLLM. Specifically, we adopt the distillation strategy of combining MDist and RDist, and we freeze the visual encoder and optimize the projector and $s$-LLM. By using MDist, the small-scale $s$-LLM in the $s$-MLLM can be fully optimized to better simulate the reasoning ability of the large scale $l$-LLM. And RDist can further facilitate the $s$-MLLM to learn the visual representation of the $l$-MLLM. Overall, the training objective can be defined as:

$$\mathcal{L}_{DFT} = \mathcal{L}_{reg} + \alpha'\mathcal{L}_{res} + \beta'\mathcal{L}_{vis} + \gamma'\mathcal{L}_{rel} \tag{7}$$

where $\mathcal{L}_{reg}$ denotes the auto-regressive prediction loss, $\alpha'$, $\beta'$ are weights for visual and response distribution in MDist, and $\gamma'$ is weight for RDist.

### 3.3.3 DISCUSSION WITH RECENT LLAVA-MOD

We compare our approach with the recently released LLaVA-MoD (Shu et al., 2024) for MLLM distillation to highlight the technical differences: *1)* In terms of training strategy, we design an additional DFT stage, whereas LLaVA-MoD introduces Preference Distillation. *2)* Structurally, we do not incorporate complex architectures for $s$-MLLM, while LLaVA-MoD employs MoE modeling. *3)* Regarding the training function, we develop KD-oriented MDist/RDist losses for the DPT and DFT stages, whereas LLaVA-MoD introduces PO Loss in the Preference Distillation stage.

## 4 EXPERIMENTAL RESULTS

### 4.1 SETUP

**Implementation Details.** For both the large/small-scale MLLMs, we employ the pre-trained SigLIP-B/14@384px (Zhai et al., 2023) as the Visual Encoder and a two-layer MLP with a GELU activation layer as the Projector, while adopting Qwen1.5 (Yang et al., 2024) family as LLM models. $l$-MLLM equipped with 4B parameters serves as the teacher model, while the $s$-MLLM is configured with 0.5B or 1.8B parameters. During training, we utilize the LLaVA1.5-558k (Liu et al., 2024a) for DPT stage, and LLaVA-mix-665k (Liu et al., 2024a) for both SFT and DFT stages. During the DPT stage, the loss weights $\alpha$, $\beta$, and $\gamma$ are set to 1.0, 1.0, and 0.5, respectively. Batch size is set to 32 and the learning rate is 1e-3. During SFT and DFT stages, the loss weights $\alpha'$, $\beta'$, and $\gamma'$ are set to 1.0, 1.0, and 0.5, and we set batch size 16 and learning rate 2e-5. We train for one epoch at each stage and utilize the AdamW optimizer (Loshchilov, 2017) with the cosine learning rate schedule for all stages. All experiments are conducted on 8 NVIDIA L40s GPUs. The entire training process for $s$-MLLMs configured with 0.5B and 1.8B parameters take approximately 210 and 320 GPU hours. The experiments are conducted based on the TinyLLaVA factory (Zhou et al., 2024).

**Details of Comparison Methods.** We primarily compare with recent efforts focused on small-scale MLLMs, including Imp (Shao et al., 2024), Bunny (He et al., 2024), Mini-Gemini (Li et al., 2024), MoE-LLaVA (Lin et al., 2024), SPHINX (Gao et al., 2024), and LLaVA-MoD (Shu et al., 2024). Additionally. we also compare our LLaVA-KD with current state-of-the-art MLLMs, such as BLIP-2 (Li et al., 2023a), Instruct-BLIP (Dai et al., 2023), mPLUG-Owl2 (Ye et al., 2024), LLaVA1.5 (Liu et al., 2024a), TinyLLaVA (Zhou et al., 2024), LLaVA-Phi (Zhu et al., 2024), MobileVLM (Chu et al., 2023), MiniCPM-V (Yao et al., 2024).

**Benchmark Settings.** General VQA requires the model to generate answers based on the image and related question, necessitating the ability to understand how visual and textual information interrelate. For general VQA, we evaluate LLaVA-KD on four benchmarks including VQAv2 (Goyal et al., 2017), GQA (Hudson & Manning, 2019), VizWiz (Gurari et al., 2018), and ScienceQA (Image set) (Lu et al., 2022). Scene Text-centric VQA (TextVQA (Singh et al., 2019)) requires the model recognize and understand textual information in an image. Additionally, we utilize five popular benchmarks for evaluation including MME (Fu et al., 2023), MMB (Liu et al., 2023), MMB$^{\text{CN}}$ (Liu et al., 2023), POPE (Li et al., 2023b), and MMMU (Yue et al., 2024).

### 4.2 BENCHMARKED RESULTS WITH THE STATE-OF-THE-ARTS

As shown in Table 1, In the context of 1B and 2B model scales, our LLaVA-KD demonstrates significant advantages. Specifically, with 1B parameters, we surpass SPHINX-Tiny (Gao et al., 2024) by 3.7% on average across nine benchmarks (excluding MMMU), using only 1M training samples compared to SPHINX-Tiny's 15M. (See Table 5 for more details) Furthermore, our model surpasses LLaVA-MoD (Shu et al., 2024), a model that mitigates hallucination through preference distillation, by achieving an average improvement of 1.1% across the seven reported benchmarks, excluding VQAv2, POPE, and MMMU. It's worth noting that LLaVA-MoD introduces a MoE structure in the $s$-MLLM, resulting in large total parameters. Meanwhile, LLaVA-MoD is trained on nearly five times the amount of data compared to our approach (Refer to Table 5). Moreover, it can be observed that our LLaVA-KD-1B achieves comparable results with recent the state-of-the-art $s$-MLLM MoE-LLaVA-2B (Lin et al., 2024) and surpasses TinyLLaVA-2B (Zhou et al., 2024), despite having only half the model size. It also can be observed that, with 2B parameters, our LLaVa-KD-2B also

Table 1: **Benchmarked results with SoTA MLLMs**. Compared with counterparts, our LLaVA-KD achieves highly competitive results than current small-scale MLLM models and the recently released LLaVA-MOD (Shu et al., 2024) that employs MoE strategies. Optimal and sub-optimal results are in **bold** and underline, respectively. grey and blue backgrounds respectively represent the most direct MLLM distillation method and our approach. AVG: The average of the nine benchmarks for comprehensive comparison except MMMU. †: reproduced results using the official code.

| Method | LLM | Image Question Answering | | | | | Benchmarks | | | | | AVG |
|---|---|---|---|---|---|---|---|---|---|---|---|---|
| | | VQAv2 | GQA | VizWiz | SciQA | TextVQA | MME | MMB | MMB$^{CN}$ | POPE | MMMU | |
| BLIP-2 | Vicuna-13B | 65.0 | 41.0 | 19.6 | 61.0 | 42.5 | 64.7 | - | - | 85.3 | 34.4 | - |
| LLaVA-NeXT | Vicuna-1.5-13B | - | 65.4 | 60.5 | 73.6 | 67.1 | 76.0 | 70.0 | 64.4 | - | - | - |
| LLaVA-1.5 | Vicuna-7B | 78.5 | 62.0 | 50.0 | 66.8 | 58.2 | 75.5 | 64.3 | 58.3 | 85.9 | 34.4 | 66.6 |
| InstructBLIP | Vicuna-7B | - | 49.2 | 34.5 | 60.5 | 50.1 | - | 36.0 | 23.7 | 79.8 | - | - |
| Qwen-VL | Qwen-7B | 78.8 | 59.3 | 35.2 | 67.1 | 63.8 | - | 38.2 | 7.4 | - | - | - |
| Qwen-VL-Chat | Qwen-7B | 78.2 | 57.5 | 38.9 | 68.2 | 61.5 | 74.4 | 60.6 | 56.7 | - | 35.9 | - |
| mPLUG-Owl2 | LLaMA2-7B | 79.4 | 56.1 | 54.5 | 68.7 | 54.3 | 72.5 | 66.5 | - | 85.8 | 32.7 | - |
| TinyLLaVA† | Qwen1.5-4B | 79.9 | 63.4 | 46.3 | 72.9 | 59 | 69.25 | 67.9 | 67.1 | 85.2 | 38.9 | 67.9 |
| TinyLLaVA | Phi2-2.7B | 79.9 | 62.0 | - | 69.1 | 59.1 | 73.2 | 66.9 | - | 86.4 | 38.4 | - |
| Bunny | Phi2-2.7B | 79.8 | 62.5 | 43.8 | 70.9 | 56.7 | 74.4 | 68.6 | 37.2 | - | 38.2 | - |
| Imp-3B | Phi2-2.7B | - | 63.5 | 54.1 | 72.8 | 59.8 | - | 72.9 | 46.7 | - | - | - |
| MobileVLM | MLLaMA-2.7B | - | 59.0 | - | 61.0 | 47.5 | 64.4 | 59.6 | - | 84.9 | - | - |
| MobileVLMv2 | MLLaMA-2.7B | - | 61.1 | - | 70 | 57.5 | 72.0 | 63.2 | - | 84.7 | 30.8 | - |
| MoE-LLaVA | Phi2-2.7B | 79.9 | 62.6 | - | 70.3 | 57.0 | - | 68.0 | - | 85.7 | - | - |
| LLaVA-Phi | Phi2-2.7B | 71.4 | - | - | 68.4 | 48.6 | 66.8 | 59.8 | - | 85.0 | - | - |
| MiniCPM-V | MiniCPM-2.4B | - | 51.5 | 50.5 | 74.4 | 56.6 | 68.9 | 64.0 | 62.7 | 79.5 | - | - |
| MiniCPMv2 | MiniCPM-2.4B | - | 52.1 | 60.2 | 76.3 | 73.2 | 70.5 | 68.5 | 67.2 | 86.3 | - | - |
| LLaVADI | MLLaMA-2.7B | - | 61.4 | - | 64.1 | 50.7 | 68.8 | 62.5 | - | 86.7 | - | |
| Imp-2B | Qwen1.5-1.8B | **79.2** | 61.9 | 39.6 | 66.1 | 54.5 | 65.2 | 63.8 | 61.3 | 86.7 | - | 64.3 |
| Bunny-2B | Qwen1.5-1.8B | 76.6 | 59.6 | 34.2 | 64.6 | 53.2 | 65.0 | 59.1 | 58.5 | 85.8 | - | 61.8 |
| Mini-Gemini-2B | Gemma-2B | - | 60.7 | 41.5 | 63.1 | 56.2 | 67.0 | 59.8 | 51.3 | 85.6 | 31.7 | - |
| MoE-LLaVA-2B | Qwen-1.5-1.8B | 76.2 | 61.5 | 32.6 | 63.1 | 48.0 | 64.6 | 59.7 | 57.3 | **87.0** | - | 61.1 |
| TinyLLaVA† | Qwen1.5-1.8B | 73.1 | 55.5 | 34.9 | 65.3 | 47.7 | 61.2 | 57.1 | 55.5 | 83.4 | **34.1** | 59.3 |
| LLaVA-MOD | Qwen1.5-1.8B | - | 58.7 | 39.2 | **68.0** | **58.5** | 66.7 | **66.3** | 61.9 | **87.0** | - | - |
| LLaVA-KD-2B | Qwen1.5-1.8B | 79.0 | **62.3** | **44.7** | 64.7 | 53.4 | **69.1** | 64.0 | **63.7** | 86.3 | 33.6 | **65.2** |
| SPHINX-Tiny | TinyLlama-1.1B | 74.7 | 58.0 | **49.2** | 21.5 | **57.8** | 63.1 | 52.3 | **56.6** | 82.2 | - | 57.3 |
| TinyLLaVA† | Qwen1.5-0.5B | 73.9 | 57.4 | 24.9 | 60.9 | 47.4 | 59.8 | 55 | 52.4 | 83.7 | **31.6** | 57.3 |
| LLaVADI | MLLaMA-1.4B | - | 55.4 | - | 56.0 | 45.3 | 58.9 | 55.0 | - | 84.7 | - | |
| LLaVA-MOD | Qwen1.5-0.5B | - | 56.2 | 31.6 | **62.8** | 53.9 | 65.3 | 58.8 | 50.4 | - | - | - |
| LLaVA-KD-1B | Qwen1.5-0.5B | **77.0** | **59.6** | 35.9 | 60.6 | 49.9 | 64.5 | **60.1** | 55.5 | 85.9 | 30.2 | **61.0** |

achieves the leading performance compared to existing small-scale MLLM models, outperforming the previous art Imp-2B (Shao et al., 2024) by 0.9%.

## 4.3 ABLATION STUDY AND ANALYSIS

**Three-Stage Training Recipe.** In Table 2, we study the influence of different training stages, reporting the average results across 10 benchmarks. Initially, we first follow the common Pre-Training (PT) and Supervised Fine-Tuning (SFT) recipe to train the small MLLM (Row1), achieving 54.7% average performance. A straightforward idea is to introduce the distillation strategy during the SFT stage (Row2). Despite some improvements, we believe the $L$-MLLM's capabilities are not fully utilized. Furthermore, incorporating DPT (Row3) with SFT improves the performance by 0.9%. This result reveals that through DPT, visual features are better projected into the text embedding space, facilitating LLM's understanding of multimodal information. Further employing DFT (Row4) significantly improves the model's capacities by 2.3%, achieving the best results on eight benchmarks. The improvement illustrates that through the DFT stage, the $S$-MLLM effectively acquired the knowledge from the $L$-MLLM, thereby significantly enhancing its understanding capabilities. However, when we remove the SFT stage, the performance significantly dropped to 55.9%, yet it still surpasses the result that is obtained using SFT for fine-tuning (55.6% vs. 55.9%). These results prove the necessity of the SFT stage and further validate the effectiveness of DFT.

Table 2: Ablation studies of different training stages. PT+SFT: adopts the general two-stage training scheme, *i.e.*, TinyLLaVA-1B (Zhou et al., 2024), we serve it as a baseline; PT+DFT: a naive framework that integrates distillation process during SFT; DPT+SFT: Validates the effectiveness of the Distilled Pre-Training stage; DPT+DFT: Validates the effectiveness of the Distilled Fine-Tuning stage; DPT+SFT+DFT: Validates the effectiveness of the three-stage training strategy.

| Training Scheme | Image Question Answering | | | | | Benchmarks | | | | | AVG |
|---|---|---|---|---|---|---|---|---|---|---|---|
| | VQAv2 | GQA | VizWiz | SciQA | TextVQA | MME | MMB | MMB$^{CN}$ | POPE | MMMU | |
| PT+SFT | 73.9 | 57.4 | 24.9 | 60.9 | 47.4 | 59.8 | 55.0 | 52.4 | 83.7 | **31.6** | 54.7 |
| PT+DFT | 75.1 | 57.0 | 29.5 | 60.9 | 49.2 | 59.6 | 57.3 | 55.0 | 85.5 | 29.6 | 55.8 |
| DPT+SFT | 74.6 | 57.8 | 28.8 | **61.2** | 49.1 | 59.9 | 56.9 | 51.6 | 84.3 | 31.4 | 55.6 |
| DPT+DFT | 75.5 | 58.0 | 27.5 | 59.7 | 49.3 | 60.6 | 57.7 | 54.7 | 85.4 | 30.3 | 55.9 |
| DPT+SFT+DFT | **77.0** | **59.6** | **35.9** | 60.6 | **49.9** | **64.5** | **60.1** | **55.5** | **85.9** | 30.2 | **57.9** |

Table 3: Ablation study on Multimodal Distillation and Relation Distillation during both the Distilled Pre-Training and Distilled Fine-Tuning stages.

| Distilled Pre-Training | | Supervised Fine-Tuning | Distilled Fine-Tuning | | AVG |
|---|---|---|---|---|---|
| Multimodal Distillation | Relation Distillation | | Multimodal Distillation | Relation Distillation | |
| ✗ | ✓ | | ✗ | ✗ | 55.5 |
| ✓ | ✗ | ✓ | ✗ | ✗ | 55.1 |
| ✓ | ✓ | | ✗ | ✗ | **55.6** |
| ✓ | ✓ | | ✗ | ✓ | 57.0 |
| ✓ | ✓ | ✓ | ✓ | ✗ | 57.7 |
| ✓ | ✓ | | ✓ | ✓ | **57.9** |

**Training Strategy.** As shown in Table 3, we explore the influence of different distillation strategies, including MDist and RDist, during both the DPT and DFT stages. First, we report the results of DPT using different distillation strategies, followed by Supervised Fine-Tuning (Rows1-4). The results show that using RDist alone is more effective than using MDist alone. We believe this is because RDist helps enhance the small MLLMs' ability to model complex visual features, thereby promoting the alignment of vision and language features. During the DFT stage, using MDist alone is more effective than using RDist alone. We speculate that this is because, at this stage, directly mimicking the output distribution of the large MLLMs can enhance the understanding and reasoning abilities of small MLLMs. In both distillation stages, combining MDist and RDist shows the best results. The results demonstrate that combining MDist and RDist helps to comprehensively transfer the knowledge from large MLLMs to small MLLMs. Please refer to Sec. A.1 for more details.

**Distillation Targets.** As shown in Table 4, we validate the effectiveness of different distillation targets during both the DPT stage and DFT stage. In these experiments, we only employ the multimodal distillation to avoid the potential impact of Relation distillation. The results indicate that, unlike most existing methods that focus solely on distilling the response, incorporating visual distillation achieves the best results, whether in the DPT or DFT stage. We believe the reason is that, in the DPT stage, adding visual constraints helps improve the quality of visual features in the small-scale MLLM, thereby promoting the alignment of visual and language information, facilitating unified

Table 4: Ablation studies on the effectiveness of different distillation targets during both the Distilled Pre-Training (DPT) and Distilled Fine-Tuning (DFT) stages.

(a) Distillation targets during the DPT stage.

| Response tokens | Prompt tokens | Visual tokens | Average |
|---|---|---|---|
| ✓ | ✗ | ✗ | 54.9 |
| ✓ | ✓ | ✗ | 55.0 |
| ✓ | ✗ | ✓ | 55.1 |
| ✓ | ✓ | ✓ | 54.6 |

(b) Distillation targets during the DFT stage.

| Response tokens | Prompt tokens | Visual tokens | Average |
|---|---|---|---|
| ✓ | ✗ | ✗ | 57.2 |
| ✓ | ✓ | ✗ | 56.9 |
| ✓ | ✗ | ✓ | 57.7 |
| ✓ | ✓ | ✓ | 57.1 |

Figure 3: Qualitative visualization comparison between our LLaVA-KD 🐨 with TinyLLaVA 🖥.

understanding by the LLM. In the DFT stage, distillation on the visual distribution further enhances the model's understanding and reasoning capabilities. Please refer to Sec. A.2 for more details.

**Efficiency comparison of SoTA MLLMs.** In Table 5, we compare our model with SoTA small-scale MLLMs in terms of model size (#Params), training samples (#Samples) and training time (Time). The "AVG" is computed on seven benchmarks, excluding VQAv2, POPE, and MMMU, for comprehensive comparison. With 1B parameters, compared to SPHINX-Tiny (Gao et al., 2024) and LLaVA-MoD (Shu et al., 2024), our LLaVA-KD outperforms them by 4.0% and 1.1%, respectively, while utilizing less training data. With 2B parameters, we can observe the similar trend. Compared to Imp (Shao et al., 2024) and LLaVA-MoD, we achieve improvements of {1.4% and 0.4%, respectively. Compared to TinyLLaVA, despite an increase in training time, LLaVA-KD achieves a significant performance improvement of 4.1% and 6.4% under the 1B and 2B parameters, respectively. Overall, our method achieves a favorable balance between training time and performance compared to existing SoTA $s$-MLLM models.

Table 5: Efficiency comparison of SoTA MLLMs.

| Method | #Params | #Samples | Time | AVG |
|---|---|---|---|---|
| TinyLLaVA | | 1.2 M | 105 | 53.9 |
| MoE-LLaVA | ~2B | 2.2 M | / | 55.3 |
| Bunny | | 2.6M | / | 56.3 |
| Mini-Gemini | | 2.7M | / | 57.1 |
| Imp | | 1.5M | / | 58.9 |
| LLaVA-MoD | | 5 M | 960 | 59.9 |
| LLaVA-KD | | 1.2 M | 320 | 60.3 |
| TinyLLaVA | | 1.2 M | 52 | 51.1 |
| SPHINX-Tiny | ~1B | 15 M | / | 51.2 |
| LLaVA-MoD | | 5 M | / | 54.1 |
| LLaVA-KD | | 1.2 M | 210 | 55.2 |

### 4.4 FURTHER VISUALIZATION AND EXPLORATION

**Visualization.** Fig. 3 shows qualitative results between our LLaVA-KD-1B and TinyLLaVA-1B (Zhou et al., 2024). It can be observed that our approach achieves a more accurate understanding of multimodal information, leading to more precise responses.

**Futher Exploration.** It should be noted that in our framework, to ensure that the $s$-MLLM can effectively learn from the $l$-MLLM, both $l$-MLLM and $s$-MLLM need to employ the same series of LLMs to maintain consistency in the vocabulary space. Future research can explore overcoming this limitation to integrate different MLLMs, thereby acquiring richer knowledge and capabilities to develop a more powerful teacher model, and further enhancing the performance of the $s$-MLLM.

## 5 CONCLUSION

This paper introduces the LLaVA-KD, a framework that transfers knowledge from a $l$-MLLM to a $s$-MLLM. This approach effectively reduces model size and computational complexity while enabling the $s$-MLLM to maintain the capabilities of the $l$-MLLM. LLaVA-KD introduces a distillation strategy, including MDist and RDist. MDist minimizes the discrepancy between the visual-textual output distributions of $l$-MLLM and $s$-MLLM. RDist transfers $l$-MLLM's capacity to model correlations between visual features. In addition, we propose a three-stage training scheme to fully exploit the potential of $s$-MLLM: DPT to promote the alignment between visual-textual features, SFT to equip model with multimodal understanding, and DFT to further transfer $l$-MLLM capacities. Comprehensive experiments on ten benchmarks demonstrate the effectiveness of our framework.

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

# A APPENDIX

## A.1 DETAILED QUANTITATIVE RESULTS ON USING DIFFERENT DISTILLATION STRATEGIES

Table A1 and Table A2 respectively present the results of adopting different distillation strategies during the Distilled Pre-Training stage and the Distilled Fine-Tuning stage.

Table A1: Detailed results of the ablation study on different distillation strategies during the Distilled Pre-Training stage.

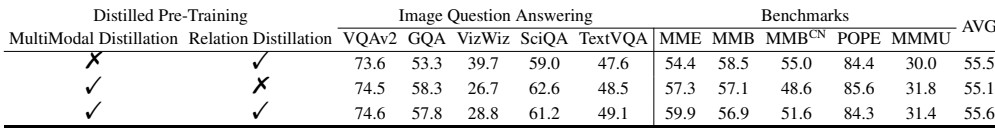

| Distilled Pre-Training | | Image Question Answering | | | | | Benchmarks | | | | | AVG |
|---|---|---|---|---|---|---|---|---|---|---|---|---|
| MultiModal Distillation | Relation Distillation | VQAv2 | GQA | VizWiz | SciQA | TextVQA | MME | MMB | MMB$^{CN}$ | POPE | MMMU | |
| ✗ | ✓ | 73.6 | 53.3 | 39.7 | 59.0 | 47.6 | 54.4 | 58.5 | 55.0 | 84.4 | 30.0 | 55.5 |
| ✓ | ✗ | 74.5 | 58.3 | 26.7 | 62.6 | 48.5 | 57.3 | 57.1 | 48.6 | 85.6 | 31.8 | 55.1 |
| ✓ | ✓ | 74.6 | 57.8 | 28.8 | 61.2 | 49.1 | 59.9 | 56.9 | 51.6 | 84.3 | 31.4 | 55.6 |

Table A2: Detailed results of the ablation study on different distillation strategies during the Distilled Fine-Tuning stage.

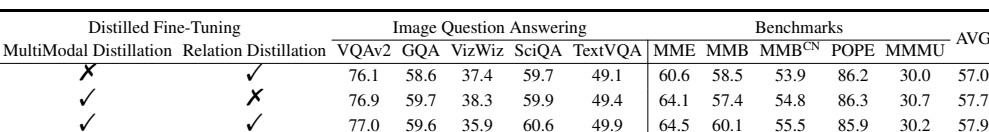

| Distilled Fine-Tuning | | Image Question Answering | | | | | Benchmarks | | | | | AVG |
|---|---|---|---|---|---|---|---|---|---|---|---|---|
| MultiModal Distillation | Relation Distillation | VQAv2 | GQA | VizWiz | SciQA | TextVQA | MME | MMB | MMB$^{CN}$ | POPE | MMMU | |
| ✗ | ✓ | 76.1 | 58.6 | 37.4 | 59.7 | 49.1 | 60.6 | 58.5 | 53.9 | 86.2 | 30.0 | 57.0 |
| ✓ | ✗ | 76.9 | 59.7 | 38.3 | 59.9 | 49.4 | 64.1 | 57.4 | 54.8 | 86.3 | 30.7 | 57.7 |
| ✓ | ✓ | 77.0 | 59.6 | 35.9 | 60.6 | 49.9 | 64.5 | 60.1 | 55.5 | 85.9 | 30.2 | 57.9 |

## A.2 DETAILED QUANTITATIVE RESULTS ON USING DIFFERENT DISTILLATION TARGETS

Table A3 and Table A4 respectively demonstrate the results of employing different distillation targets during the Distilled Pre-Training stage and the Distilled Fine-Tuning stage.

Table A3: Detailed results of the ablation study on the different distillation targets during the Distilled Pre-Training stage.

| Response Tokens | Prompt Tokens | Visual Tokens | Image Question Answering | | | | | Benchmarks | | | | | AVG |
|---|---|---|---|---|---|---|---|---|---|---|---|---|---|
| | | | VQAv2 | GQA | VizWiz | SciQA | TextVQA | MME | MMB | MMB$^{CN}$ | POPE | MMMU | |
| ✓ | ✗ | ✗ | 73.8 | 57.8 | 25.6 | 62.8 | 47.1 | 59.7 | 55.9 | 49.3 | 85.5 | 31.6 | 54.9 |
| ✓ | ✓ | ✗ | 74.1 | 58.2 | 24.4 | 60.6 | 48.6 | 59.9 | 56.3 | 50.6 | 84.8 | 32.3 | 55.0 |
| ✓ | ✗ | ✓ | 74.5 | 58.3 | 26.7 | 62.6 | 48.5 | 57.3 | 57.1 | 48.6 | 85.6 | 31.8 | 55.1 |
| ✓ | ✓ | ✓ | 74.2 | 58.3 | 24.6 | 60.4 | 46.9 | 60.0 | 55.6 | 49.1 | 84.8 | 32.2 | 54.6 |

Table A4: Detailed results of the ablation study on the different distillation targets during the Distilled Fine-Tuning stage.

| Response Tokens | Prompt Tokens | Visual Tokens | Image Question Answering | | | | | Benchmarks | | | | | AVG |
|---|---|---|---|---|---|---|---|---|---|---|---|---|---|
| | | | VQAv2 | GQA | VizWiz | SciQA | TextVQA | MME | MMB | MMB$^{CN}$ | POPE | MMMU | |
| ✓ | ✗ | ✗ | 76.8 | 59.6 | 36.4 | 59.1 | 50.2 | 64.0 | 57.6 | 52.7 | 85.8 | 30.1 | 57.2 |
| ✓ | ✓ | ✗ | 77.0 | 59.5 | 27.5 | 60.1 | 51.5 | 62.7 | 59.5 | 55.8 | 85.7 | 30.0 | 56.9 |
| ✓ | ✗ | ✓ | 76.9 | 59.7 | 38.3 | 59.9 | 49.4 | 64.1 | 57.4 | 54.8 | 86.3 | 30.7 | 57.7 |
| ✓ | ✓ | ✓ | 76.4 | 59.0 | 30.8 | 61.4 | 49.9 | 63.5 | 59.2 | 55.1 | 86.0 | 29.9 | 57.1 |

