# OpenReview forum: "A Framework of Distilling Multimodal Large Language Models"
_ICLR.cc/2025/Conference — Submitted to ICLR 2025_

### Official Review · Reviewer_Qt7K · 2024-10-28

**Soundness:** 3
**Presentation:** 3
**Contribution:** 3
**Rating:** 6
**Confidence:** 3

**Summary:**

This paper introduces a three-stage pipeline to distill knowledge from a large MLLM to a smaller one. The proposed distillation strategy, particularly RDist, is both innovative and effective in transferring the capabilities of the large MLLM. Experiments across 10 benchmarks further validate the effectiveness of this method.

**Strengths:**

1. This paper extends the common PT-SFT two-stage training scheme to a three-stage training strategy, enhancing the multimodal alignment process and transferring the capacities from the large MLLM to the smaller one.
2. The proposed Multimodal Distillation (MDist) and Relational Distillation (RDist) are novel. The RDist seems sound to learn the visual representation of the l-MLLM.
3. The performance of LLaVA-KD is quite impressive, significantly outperforming the baseline method.

**Weaknesses:**

1. Some notations are confusing, e.g.,
    1. In Eq. (1), $\phi_l(y_m | y_{<m})$ seems to be a probability, but the author claims that it is a distribution.
    2. In Eq. (2), $\phi_l(Y_j | y_{<m})$ is a probability, but \phi_l(y_m | y_{<m}) is a distribution.
    3. What is $\mathcal{L}_{PT}$ in Eq. (6)?
2. From Table.2, it seems that PT + DFT is comparable to DPT + DFT.  Can you provide the result of PT+SFT+DFT to further validate the effectiveness of DPT?
3. (Minor) In Section 3.3, “we further design a three-stage distillation scheme (Sec. 3.3.1)”, (Sec. 3.3.1) should be (Sec. 3.3.2).

**Questions:**

See weaknesses.

---

> ### Author Response · Authors · 2024-11-23
> **Response to Reviewer Qt7K**
>
> We sincerely appreciate your constructive comments and suggestions. Thank you for your efforts in assessing and helping to improve the quality of our manuscripts. Below are our responses to your concerns:
>
> **Q4.1 Some notations are confusing**
> 1. In Eq.(1), $\phi_l (y_{m} \mid \mathbf{y}_{<m})$ is a conditional probability distribution that describes the distribution of the random variable $y_m$ given the previous observations;
>
> 2. Your understanding is correct
> - $\phi_{l}\left(Y_{j} \mid \mathbf{y}_{<m}\right)$ is a probability.
> - $\phi_{s}(y_m \mid \mathbf{y}_{<m})$ is a distribution.
>
> For discrete distributions $P$ and $Q$, the definition of KLD is $\text{KLD}(P | Q)=\sum_{i} P(i) \log \left(\frac{P(i)}{Q(i)}\right)$.
> The Eq.(2)  indicates that when calculating KLD, we sum the probabilities of each word in the vocabulary, where
> - $\phi_{l}\left(Y_{j} \mid \mathbf{y}_{<m}\right)$ represent the probability of the $j$-th word $Y_j$ in the vocabulary at position $m$ for the teacher model,
> - $\phi_{s}\left(Y_{j} \mid \mathbf{y}_{<m}\right)$ represent the probability of the $j$-th word $Y_j$ in the vocabulary at position $m$ for the student model.
>
> 3. Thank you for your review. $L_{PT}$ should be revised to $L_{reg}$ in Eq.(1), representing the autoregressive training objective. We have made this correction in the revised version.
>
> **Q4.2 Can you provide the result of PT+SFT+DFT to further validate the effectiveness of DPT?**
>
> We additionally supplement the experiments with PT+SFT+DFT, and the results demonstrate that DPT+SFT+DFT surpasses PT+SFT+DFT by 1.3%, proving the effectiveness of the DPT stage.
>
> The equivalence in final performance between DPT+DFT and PT+DFT may be attributed to the fact that the distillation process in the DFT stage plays a crucial role in both combinations, bringing the model to a similar level of comprehension ability. Specifically, although DPT provides better multimodal mapping in the initial stage, the powerful capability of DFT compensates for the shortcomings of the PT stage, resulting in a final performance comparable to DPT+DFT and PT+DFT.
>
> Furthermore, the performance of DPT+SFT is significantly better than PT+SFT (Table 2 in the subscription), which demonstrates that DPT, compared to PT, is more conducive to MLLMs achieving the mapping from the visual space to the text space. Consequently, in the SFT stage, without the guidance of a teacher network, s-MLLM can optimize from a higher baseline, enhancing the model's final performance.
>
> | method      | VQAv2                    | GQA  | VisWiz | SciQA | TextVQA | MME  | MMB | $\text{MMB}^{\text{CN}}$  | POPE | MMMU |  Avg      |
> |:-----------:|:------------------------:|:----:|:------:|:-----:|:-------:|:----------:|:-----------:|:----------:|:----:|:---------:|:--------:|
> | PT+SFT+DFT  | 76.6                     | 59.4 | 32.6   | 60.4  | 48.4    | 60.9       | 57.8        | 54         | 84.9 | 31.3      | 56.6  |
> | DPT+SFT+DFT | 77.0                     | 59.6 | 35.9   | 60.6  | 49.9    | 64.5       | 60.1        | 55.5       | 85.9 | 30.2      | 57.9    |
>
> **Q4.3 In Section 3.3, “we further design a three-stage distillation scheme (Sec. 3.3.1)”, (Sec. 3.3.1) should be (Sec. 3.3.2).**
>
> Thank you for your careful review. We have made corrections to the citations in the revised paper.

---

> > ### Comment · Reviewer_Qt7K · 2024-11-27
> >
> > Thank you for the detailed responses that address my concerns. I will maintain my score to accept this paper.

---

> > > ### Author Response · Authors · 2024-11-27
> > >
> > > Dear Reviewer Qt7K,
> > >
> > > Thank you for your positive and encouraging feedback on our work! We’re delighted to hear that our rebuttal has addressed your concerns. We sincerely appreciate the time and effort you’ve invested in providing detailed reviews and valuable suggestions to help improve our work.
> > >
> > > Best regards,
> > >
> > > Authors of LLaVA-KD #686

---

### Official Review · Reviewer_LdWi · 2024-11-03

**Soundness:** 3
**Presentation:** 3
**Contribution:** 2
**Rating:** 6
**Confidence:** 3

**Summary:**

Distillation techniques are commonly employed to effectively train small models using given large models. While numerous studies have explored training small-scale LLMs through distillation, there has been limited work on distillation methods for MLLMs. To address this gap, this paper presents various training strategies for MLLM distillation.

1. This paper proposes a three-stage distillation scheme: i) Distilled Pre-Training (DPT), ii) Supervised Fine-Tuning (SFT), and iii) Distilled Fine-Tuning (DFT).
2. This paper proposes two distillation loss terms: i) Multimodal Distillation (MDist) and ii) Relational Distillation (RDist), which are tailored for multimodality.

**Strengths:**

1. The paper is well-motivated: This paper is timely given that while the field of MLLMs is rapidly evolving, research on distillation methods for MLLMs remains largely unexplored.
2. The paper is well-written and easy to follow.
3. The proposed LLaVA-KD models demonstrate superior performance compared to existing small-scale MLLMs.

**Weaknesses:**

1. It appears that the methods proposed in this paper (three-stage distillation scheme, Multimodal Distillation) do not present particularly novel or distinctive contributions to the field. Therefore, systematic experimentation would be essential to demonstrate clear superiority of the proposed methods.
2. The discussion and comparison with existing distillation methods for MLLMs appears insufficient.
    * While the paper cites LLaVA-MoD [1] as related work, it fails to conduct comparative experiments against the Preference Distillation technique proposed in LLaVA-MoD. The performance comparison between LLaVA-MoD and LLaVA-KD models alone cannot provide meaningful insights due to the lack of controlled variables.
    * Additionally, the paper lacks discussion and comparative experiments with LLAVADI [2], another concurrent work on MLLM distillation.
3. The experimental settings appear inadequate for demonstrating the effectiveness of the proposed method.
    * While this paper focuses on performance comparisons with existing small-scale MLLMs, it appears difficult to claim the superiority of the proposed distillation techniques solely through these comparisons. A direct performance comparison with existing LLM/MLLM distillation methods under the same experimental settings would be more convincing.
    * Please refer to the Questions section for additional details.

[1] LLaVA-MoD: Making LLaVA Tiny via MoE Knowledge Distillation
[2] LLAVADI: What Matters For Multimodal Large Language Models Distillation

**Questions:**

1. Why does this paper emphasize that LLaVA-KD's performance is better than BLIP series? Since this is a paper presenting a distillation method, directly comparing performance between different types of models seems inappropriate. Rather, it would be meaningful to report the performance of small models after applying the proposed distillation method to the BLIP series.
2. Can we directly compare the performance of the proposed method with the distillation methods presented in LLaVA-MoD and LLAVDI under the same experimental settings?
3. How does the performance compare when directly applying the LLM distillation methods mentioned in Section 2.2's 'KD for LLM' paragraph to MLLM distillation? A comparative analysis would be necessary to determine whether MLLM-tailored distillation methods are truly needed or if existing LLM distillation methods are sufficient.
4. What is the performance of PT+SFT+DFT? The comparable performance between PT+DFT and DPT+DFT raises questions about the necessity of DPT over PT.
5. A study on hyperparameters (loss weights) seems necessary. The importance of hyperparameter tuning for effective distillation should be investigated.

---

> ### Author Response · Authors · 2024-11-23
> **Response to Reviewer LdWi (1/4)**
>
> We sincerely appreciate your constructive comments and suggestions. Thank you for your efforts in assessing and helping to improve the quality of our manuscripts. Below are our responses to your concerns:
>
> **Q3.1 It appears that the methods do not present particularly novel or distinctive contributions to the field. Therefore, systematic experimentation would be essential to demonstrate clear superiority of the proposed methods.**
>
> Thank you for your careful review. We hope to further clarify our motivation and the comparison with other methods:
> - **Motivation:**
> Knowledge distillation has not been fully explored in the training of $s$-MLLMs. Therefore, we believe that exploring effective training strategies based on knowledge distillation is timely and necessary, providing new ideas for the community to develop small models.
>
> - **Comparison with Existing Work:**
>    - Some existing works have already explored knowledge distillation in MLLMs. For example, LLaVADI experimentally verified that using teacher models for data augmentation is beneficial for enhancing the capabilities of s-MLLMs; LLaVA-MoD focused on improving the structure of s-MLLMs and aimed to address the hallucination ability of the models.
>    - Our approach focuses on improving the training strategy. We base our method on the most common training data, without the need for additional data augmentation or complex model structure design, aiming to enhance the performance of s-MLLMs through improvements in the training strategy.
>
> - **Experimental Improvements:**
> To better demonstrate the superiority of our method, we have further supplemented and refined the experimental section according to your suggestions. Please refer to the subsequent responses for detailed information.
>
> **Q3.2 It fails to conduct comparative experiments against the Preference Distillation technique proposed in LLaVA-MoD.**
>
> Thank you for your comments.
> To address your issue, we replace the DFT stage in our DPT+SFT+DFT training stages with **Mimic Distillation (MD)** and **Preference Distillation(PD)**. As illustrated in the following table, when employing the DPT+SFT+MD strategy, our method demonstrates an average improvement of 1.0%. However, when PD is additionally incorporated, the performance decreases to 55.1%, with our method showing an advantage of 2.8%.
>
> Our finding is consistent with LLaVA-MoD: "Preference distillation remarkably reduces the hallucination of s-MLLM, while it does not yield consistent improvements in comprehension capability."  It appears that PD primarily addresses the model's hallucination ability, but this also results in a decline in the model's comprehension capability.
>
> |                              | VQAv2                    | GQA  | VisWiz | SciQA | TextVQA | MME  | MMB | $\text{MMB}^{\text{CN}}$  | POPE | MMMU | Avg    |
> |------------------------------|--------------------------|------|--------|-------|---------|------------|-------------|------------|------|-----------|--------|
> | DPT+SFT+MD | 76.3                     | 58.5 | 31.6   | 58.4  | 51.7    | 60.6       | 59.6        | 55.8       | 86.2 | 30.2      | 56.9  |
> | DPT+SFT+MD+PD | 74.4	|57.1	 | 22.7	| 58.4	| 47.7	| 58.4 |	58.8	| 54.5	| 86.6	| 32.1 | 55.1 |
> | DPT+SFT+DFT      (Ours)             | 77.0                     | 59.6 | 35.9   | 60.6  | 49.9    | 64.5     | 60.1       | 55.5       | 85.9 | 30.2      | 57.9  |
>
>
>
>
> **Q3.3 Additionally, the paper lacks discussion and comparative experiments with LLAVADI [2], another concurrent work on MLLM distillation.**
>
> Thank you for the valuable comments. We have discussed LLaVADI in the related work section of the revised paper and added a comparison with LLaVADI in Table 1.
>
> Regarding the experimental comparison, we would like to state that since LLaVADI has not been open-sourced and due to the lack of some experimental details (such as the hyperparameters of the weights of different training objectives during training, the source of data when using teacher models for data augmentation, etc.), it is quite difficult to reproduce their results. Therefore, we only compare the final performance of the two models.
>
> We report the average of the six benchmarks reported in LLaVADI. It can be seen that our 2B model outperforms their 3B model by a margin of 0.9%; our advantage is more pronounced with 1B parameters, showing a margin of 4.2%.
>
> | Method   | LLM             |  GQA  | SciQA | TextVQA | MME  | MMB | POPE  | AVG   |
> |:--------:|:---------------:|:----:|:-----:|:-------:|:----------:|:-----------:|:----:|:-----:|
> | LLaVADI  | MobileLLaMA2.7B | 61.4 | 64.1  | 50.7    | 68.8       | 62.5        | 86.7 | 65.7  |
> | LLaVA-KD (Ours) | Qwen1.5-1.8B    | 62.3 | 64.7  | 53.4    | 69.1       | 64.0          | 86.3 | 66.6  |
> | LLaVADI  | MobileLLaMA1.4B | 55.4 | 56.0  | 45.3    | 58.9       | 55.0          | 84.7 | 59.2  |
> | LLaVA-KD (Ours) | Qwen1.5-0.5B    | 59.6 | 60.6  | 49.9    | 64.5       | 60.1        | 85.9 | 63.4  |

---

> ### Author Response · Authors · 2024-11-23
> **Response to Reviewer LdWi (2/4)**
>
> **Q3.4 A direct performance comparison with existing LLM/MLLM distillation methods under the same experimental settings would be more convincing.**
>
> It is indeed necessary to validate the effectiveness of our distillation strategy under the same experimental settings. To be more convincing, we have conducted direct comparisons with existing LLM and MLLM distillation methods under the same experimental settings. Specifically:
> - For the comparison with MLLM distillation methods, please refer to our response for Q3.6.
> - For the comparison with LLM distillation methods, please refer to our response for  Q3.7.
>
> We hope these additional experiments and comparisons can furthur demonstrate the effectiveness and advantages of our method. Thank you again for your suggestions.
>
> **Q3.5 Why does this paper emphasize that LLaVA-KD's performance is better than BLIP series?**
>
> Thank you for your comments. According to your suggestions and Q2.2 raised by Reviewer sLGE, we agree that it is unfair to compare with the BLIP series models under the condition of using higher quality data for training. Therefore, we have removed the statement regarding the comparison with BLIP2 and revised the relevant parts in the revised paper.
>
> Moreover, we would like to emphasize that, compared to recent advancements in some s-MLLMs, including Imp, Moe-LLaVA, LLaVA-MoD, etc., LLaVA-KD still demonstrates superiority.
>
> **Q3.6 Can we directly compare the performance of the proposed method with the distillation methods presented in LLaVA-MoD and LLAVDI under the same experimental settings?**
>
> **LLaVA-MoD:** To compare with the distillation method proposed by LLaVA-MoD under the same experimental settings, we first attempt to replace the DFT stage in the DPT+SFT+DFT training stages with mimic Distillation and preference distillation. As shown in the following table, when we replace our DFT with Mimic Distillation, LLaVA-KD achieves a 1.0% improvement.
> The additional introduction of the preference distillation training stage results in a further decline in performance, which is consistent with the description in the original LLaVA-MoD paper:"Preference distillation remarkably reduces the hallucination of $s$-MLLM, while it does not yield consistent improvements in comprehension capability."  In this way，LLaVA-KD achieves a 2.8% improvement. These results further demonstrate the effectiveness of our distillation algorithms.
>
> |                              | VQAv2                    | GQA  | VisWiz | SciQA | TextVQA | MME  | MMB | $\text{MMB}^{\text{CN}}$  | POPE | MMMU | Avg    |
> |------------------------------|--------------------------|------|--------|-------|---------|------------|-------------|------------|------|-----------|--------|
> | DPT+SFT+Mimic Distillation  | 76.3                     | 58.5 | 31.6   | 58.4  | 51.7    | 60.6       | 59.6        | 55.8       | 86.2 | 30.2      | 56.9  |
> | DPT+SFT+Mimic Distillation+Preference Distillation  | 74.4	|57.1	 | 22.7	| 58.4	| 47.7	| 58.4 |	58.8	| 54.5	| 86.6	| 32.1 | 55.1 |
> | DPT+SFT+DFT      (Ours)             | 77.0                     | 59.6 | 35.9   | 60.6  | 49.9    | 64.5     | 60.1       | 55.5       | 85.9 | 30.2      | 57.9  |
>
> Additionally, we also transfer the Pre-Training (PT) + Mimic Distillation strategy of LLaVA-MoD to our LLaVA-KD method. As shown in the following table, it can be seen that our proposed training strategy demonstrates significant advantages.
>
> | Method     |  VQAv2                    | GQA  | VisWiz | SciQA | TextVQA | MME  | MMB | $\text{MMB}^{\text{CN}}$  | POPE | MMMU | Avg     |
> |------------|--------------------------|------|--------|-------|---------|------------|-------------|------------|------|-----------|---------|
> | PT + Mimic Distillation | 73.4                     | 57.6 | 25.6   | 59.3  | 47.2    | 59.46      | 54.9        | 49.1       | 84.6 | 32.7      | 54.4    |
> | DPT + SFT + DFT  (Ours)     | 77.0                       | 59.6 | 35.9   | 60.6  | 49.9    | 64.5       | 60.1        | 55.5       | 85.9 | 30.2      | 57.9    |
>
> **LLaVADI:** Since this paper has not been open-sourced and some important experimental details, such as loss weights and how the teacher model generates instruction-tuning data, have not been provided, reproducing this method presents significant difficulties.
> We hope the reviewers understand that our research focuses on exploring suitable training strategies for MLLMs, and we have conducted a fair comparison with the latest method, LLaVA-MoD, under the same experimental settings, demonstrating the superiority of our distillation algorithms and training strategies.

---

> ### Author Response · Authors · 2024-11-23
> **Response to Reviewer LdWi (3/4)**
>
> **Q3.7 How does the performance compare when directly applying the LLM distillation methods mentioned in Section 2.2's 'KD for LLM' paragraph to MLLM distillation?**
>
>  In LLaVA-KD, we focus on discussing the impact of distillation algorithms and training strategies on s-MLLMs and propose a three-stage training strategy along with the MDist and RDist distillation algorithms. Existing LLM training strategies mainly focus on studying the application of standard KL loss in LLM distillation and improving it. Referring to previous work LLaVADI, which also explores the application of LLM distillation algorithms in MLLMs, we followed the training strategy of  PT + DFT (Distilling Response logits with Standard KL and Reverse KL loss [1]).
>
> As shown in the following table, we find that the advancements in LLMs seem less applicable to MLLM distillation. This finding is consistent with the conclusion of LLaVADI.
>
> At the same time, by comparing LLaVA-KD's PT + DFT (Row 2) with the above results, LLaVA-KD demonstrates superior performance on 9 benchmarks, with an average advantage of 0.8. This also reveals the effectiveness of our proposed MDist+RDist distillation algorithms in MLLM distillation.
>
> Finally, adopting our three-stage training strategy achieves the best performance, which highlights the effectiveness of the training strategies we explored.
>
> |                      | VQAv2                    | GQA  | VisWiz | SciQA | TextVQA | MME  | MMB | $\text{MMB}^{\text{CN}}$  | POPE | MMMU |  Avg      |
> |----------------------|--------------------------|------|--------|-------|---------|------------|-------------|------------|------|-----------|----------|
> | PT + DFT (with LLM's strategy)          | 74.3                     | 56.6 | 26.7   | 60.8  | 49.1    | 57.8     | 56.8        | 53.7       | 84.7 | 30.0        | 55.0     |
> | PT + DFT (with MDist+RDist) | 75.1                     | 57.0   | 29.5   | 60.9  | 49.2    | 59.6       | 57.3        | 55.0    | 85.5 | 29.6      | 55.8     |
> | LLaVA-KD (Ours)             | 77.0                       | 59.6 | 35.9   | 60.6  | 49.9    | 64.5       | 60.1        | 55.5       | 85.9 | 30.2      | 57.9     |
>
> **Q3.8 What is the performance of PT+SFT+DFT? The comparable performance between PT+DFT and DPT+DFT raises questions about the necessity of DPT over PT.**
>
> The following table compares the performance of DPT+SFT+DFT with PT+SFT+DFT. We can observe that DPT+SFT+DFT surpasses PT+SFT+DFT by 1.3%.
>
> Regarding the equivalence in final performance between DPT+DFT and PT+DFT, we believe this may be attributed to the distillation process in the DFT stage. This stage plays a crucial role in both combinations, bringing the model to a similar level of comprehension ability. Specifically, although DPT provides better multimodal mapping in the initial stage, the powerful capability of DFT compensates for the shortcomings of the PT stage, resulting in a final performance comparable to DPT+DFT.
>
> Secondly, we notice that the performance of DPT+SFT is significantly better than PT+SFT (Table 2 in the subscription). This demonstrates that DPT, compared to PT, is more conducive to MLLMs achieving the mapping from the visual space to the text space. Consequently, in the SFT stage, without the guidance of a teacher network, s-MLLM can optimize from an elevated baseline, enhancing the model's final performance.
>
> | method      | VQAv2                    | GQA  | VisWiz | SciQA | TextVQA | MME  | MMB | $\text{MMB}^{\text{CN}}$  | POPE | MMMU |  Avg      |
> |:-----------:|:------------------------:|:----:|:------:|:-----:|:-------:|:----------:|:-----------:|:----------:|:----:|:---------:|:--------:|
> | PT+SFT+DFT  | 76.6                     | 59.4 | 32.6   | 60.4  | 48.4    | 60.9       | 57.8        | 54.0         | 84.9 | 31.3      | 56.6  |
> | DPT+SFT+DFT | 77.0                     | 59.6 | 35.9   | 60.6  | 49.9    | 64.5       | 60.1        | 55.5       | 85.9 | 30.2      | 57.9   |
>
> \[1\] Gu, Yuxian, et al. "MiniLLM: Knowledge distillation of large language models." The Twelfth International Conference on Learning Representations. 2024.

---

> ### Author Response · Authors · 2024-11-23
> **Response to Reviewer LdWi (4/4)**
>
> **Q3.9 A study on hyperparameters (loss weights) seems necessary.**
>
> In our study, the hyperparameters mainly involve the weights of $L_{res}$, $L_{vis}$, and $L_{rel}$ in the training objectives $L_{DPT}$ and $L_{DFT}$ during the DPT and DFT stages, including ${\alpha, \beta, \gamma}$ during DPT and ${\alpha^\prime, \beta^\prime, \gamma^\prime}$ during DFT. We default to set $\alpha=\alpha^\prime=1.0$, $\beta=\beta^\prime=1.0$, and $\gamma=\gamma^\prime=0.5$.
>
> To further verify the impact of hyperparameter adjustments on the model, we conducted several sets of hyperparameter combination experiments. As shown in the following table, using different hyperparameter combinations, the final experimental results show little difference in average performance. This indicates that our model is robust to the choice of hyperparameters.
>
> We would like to state that our goal is not to find the optimal hyperparameters, but rather to use unified parameters to ensure the robustness of the model. However, this can be a direction for future improvement, such as adaptively balancing the weights to achieve the optimal model results.
>
> | ${\alpha, \beta, \gamma}$ during DPT | ${\alpha^\prime, \beta^\prime, \gamma^\prime}$ during DFT | VQAv2                    | GQA  | VisWiz | SciQA | TextVQA | MME  | MMB | $\text{MMB}^{\text{CN}}$  | POPE | MMMU    | AVG      |
> |------------------------------------|------------------------------------|--------------------------|------|--------|-------|---------|------------|-------------|------------|------|-----------|----------|
> | {1,1, 0.5}                         | {1,1, 0.5}                         | 77.0                       | 59.6 | 35.9   | 60.6  | 49.9    | 64.5       | 60.1        | 55.5       | 85.9 | 30.2      | 57.9    |
> | {1,1, 5}                           | {1, 0.5, 0.5}                      | 76.7                     | 59.5 | 36.6   | 59.3  | 51.1    | 63.9     | 58.2        | 55.1       | 85.9 | 31.3      | 57.8    |
> | {1,1, 5}                           | {1,1, 5}                           | 76.8                     | 59.8 | 35.7   | 60.4  | 50.9    | 62.1       | 59.4        | 54.7       | 86.0   | 31.7      | 57.8    |
>
> ---

---

> > ### Comment · Reviewer_LdWi · 2024-11-26
> >
> > Thank you for your detailed response. Initially, I understood from Q3.2 that conducting comparison experiments with the Preference Distillation technique proposed in LLaVA-MoD was not feasible due to computational resource constraints. However, it seems that you eventually managed to carry out the experiments. Could you clarify how you overcame the computational resource limitations?

---

> > > ### Author Response · Authors · 2024-11-26
> > >
> > > Thank you for your reply. Initially, while debugging on a single 48G GPU, we found that the training of Preference Distillation phase required more than 48G of memory. However, in order to better address your concerns, we urgently requested and successfully obtained 8 H20s (96G each), which allowed us to successfully complete the experiment.

---

> > > > ### Comment · Reviewer_LdWi · 2024-11-26
> > > >
> > > > Thank you for your clarification. Your responses have sufficiently addressed most of my concerns, particularly Q3.6 (comparison with MLLM distillation methods). However, I still have some concerns regarding Q3.7 (comparison with LLM distillation methods). It seems this paper needs to clearly explain why a tailored distillation method for MLLMs is necessary, rather than relying on existing LLM methods. From the table, it seems the results are compared only with reverse KLD. Could you clarify if comparisons between PT + DFT (with MDist+RDist) and other simple baselines, such as PT + DFT (forward KLD) or PT + DFT (JSD), were also conducted?

---

> > > > > ### Author Response · Authors · 2024-11-26
> > > > >
> > > > > Thank you for your response. We fully understand your concerns. Due to time limitations, we previously only conducted comparisons with Reverse KLD. We will immediately conduct comparative experiments with PT + DFT (forward KLD) and PT + DFT (JSD) to verify the performance differences between our method and other LLM distillation methods. The specific results will be reported after the experiments are completed. Thank you again for your valuable feedback.

---

> > > > > ### Author Response · Authors · 2024-11-27
> > > > >
> > > > > We supplemented the experiments of PT + DFT (FKL) and PT + DFT (JSD). As shown in the following table, our MDist+RDist method achieves the best results on 7 benchmarks, demonstrating an average improvement of 0.3% compared to PT + DFT (FKL) and PT + DFT (JSD).
> > > > >
> > > > > Unlike existing LLM distillation methods, which primarily focus on optimizing the model for textual information, our distillation strategy incorporates the role of visual information in multimodal understanding tasks. This approach enables small-scale MLLMs ($s$-MLLM) to better capture the visual responses of large-scale MLLMs. Consequently, the $s$-MLLM can make more accurate predictions based on the optimized visual tokens, thereby enhancing the performance of the model.
> > > > >
> > > > >
> > > > > |                      | VQAv2                    | GQA  | VisWiz | SciQA | TextVQA | MME  | MMB | $\text{MMB}^{\text{CN}}$  | POPE | MMMU |  Avg      |
> > > > > |----------------------|--------------------------|------|--------|-------|---------|------------|-------------|------------|------|-----------|----------|
> > > > > | PT + DFT (with FKL)          |74.3	|56.1	|31.7	|59.4	|49.0	|58.9	|57.4	|54.0	|84.4	|**29.8**	|55.5 |
> > > > > | PT + DFT (with JSD)          |73.8	|54.9	|**32.3**	|60.3	|48.7	|57.6	|**57.8**	|54.3	|85.1	|**29.8**	|55.5 |
> > > > > | PT + DFT (with MDist+RDist) | **75.1**                     | **57.0**   | 29.5   | **60.9**  | **49.2**    | **59.6**       | 57.3        | **55.0**    | **85.5** | 29.6      | **55.8** |

---

> > > > > > ### Comment · Reviewer_LdWi · 2024-11-27
> > > > > >
> > > > > > Thank you for your timely clarification. I would like to acknowledge the authors for promptly conducting many additional experiments to address my concerns. While this paper could have been stronger with more systematic benchmark results—which seem challenging to complete within the remaining discussion period due to limited computation resources—I believe it serves as a solid starting point in the relatively unexplored area of MLLM distillation. Accordingly, I have decided to raise my score from 5 to 6. Thank you for your valuable work.

---

> > > > > > > ### Author Response · Authors · 2024-11-27
> > > > > > >
> > > > > > > Dear Reviewer LdWi,
> > > > > > >
> > > > > > > Thank you for your positive and encouraging feedback on our work! We’re delighted to hear that our rebuttal has addressed your concerns. We sincerely appreciate the time and effort you’ve invested in providing detailed reviews and valuable suggestions to help improve our work.
> > > > > > >
> > > > > > > Best regards,
> > > > > > >
> > > > > > > Authors of LLaVA-KD #686

---

### Official Review · Reviewer_sLGE · 2024-11-04

**Soundness:** 2
**Presentation:** 3
**Contribution:** 2
**Rating:** 3
**Confidence:** 4

**Summary:**

This paper investigates knowledge distillation for multi-modal language models (MLLMs). The proposed method introduces learning objectives targeting both visual features and language responses, designed to enhance alignment between large and small MLLMs. Additionally, the authors introduce a novel relational correlation loss that operates on the similarity matrix of visual tokens.

**Strengths:**

- The paper addresses an important and timely aspect of MLLMs, specifically focusing on the optimization of small MLLMs.
- The proposed method is clearly presented and well-explained.
- Overall, the paper is well-structured, making it clear and easy to follow.

**Weaknesses:**

In the final paragraph of the Introduction, the authors list three major contributions. However, I am not fully convinced by these claims:

- The proposed relational distillation approach lacks sufficient motivation. The rationale behind optimizing the similarity matrix of visual tokens between large and small MLLMs is unclear.
- The authors claim that their 1B model outperforms the BLIP-2 (13B) and InstructBLIP (7B) models across various benchmarks. However, this claim is misleading. BLIP-2 models are primarily trained on image-text pairs, and InstructBLIP has been trained on lower-quality instruction data compared to the extensive, high-quality LLaVA-665K dataset used in this submission. In reality, the performance difference between LLaVA and InstructBLIP is largely due to differences in data quality and quantity (besides using different ViTs). A fair comparison would require training the BLIP series models on the same dataset. Therefore, the authors' performance claims are misleading.

Regarding the proposed method, while it is straightforward, it also feels somewhat incremental, as it primarily adds a KD learning objective to various features within MLLMs.

The experimental studies and results are not convincing either:

- The performance of LLaVA-KD does not surpass that of LLaVA-MOD. In Table 1, the authors do not report the average (AVG) performance of LLaVA-MOD and omit some of its results on certain benchmarks. However, based on the reported benchmarks, the AVG performance of LLaVA-KD and LLaVA-MOD is very close.
- Additionally, the large MLLM used in this study has only 4B parameters, which may be insufficient to draw conclusions about scaling. The authors do not present its detailed performance, nor do they investigate the effect of varying large model sizes (e.g., 7B, 13B), leaving questions about scalability unaddressed.
- It is also unclear whether distilling a 1B model to another 1B model would be effective within this framework.
- The authors do not report results for small MLLMs (0.5B and 1B) trained with the standard LLaVA procedure, making it difficult to assess the improvement contributed by the proposed method.

**Questions:**

N/A

---

> ### Author Response · Authors · 2024-11-23
> **Response to Reviewer sLGE (1/2)**
>
> We sincerely appreciate your constructive comments and suggestions. Thank you for your efforts in assessing and helping to improve the quality of our manuscripts. Below are our responses to your concerns:
>
> **Q2.1  The rationale behind optimizing the similarity matrix of visual tokens between large and small MLLMs is unclear.**
>
> Relation Distillation of visual tokens is aimed at facilitating $s$-MLLMs to learn the responses of $l$-MLLMs to visual tokens, thereby enhancing the s-MLLMs' understanding of multimodal information.
>
> Our motivation is based on the observation that in order to fully understand multimodal information, MLLMs need to capture the complex relationships between different visual tokens in an image. For example, identifying the relationship of a cat lying under a table in an image, or the positional relationships between person. These all require the model to recognize and understand the interactions between visual tokens.
>
> By optimizing the visual token similarity matrix of the small model, we can explicitly transfer the relational information captured by the large model to the small model. This process enhances the small model's ability to process visual information, thereby improving its performance in multimodal tasks.
>
> **Q2.2  The authors' performance claims are misleading.**
>
> Thank you for your suggestions and viewpoints. We agree that it is unfair to compare with the BLIP series models when trained with higher quality data. Therefore, we have removed the statements regarding the comparison with BLIP2 and made corresponding modifications in the relevant sections.
>
> In addition, we would like to emphasize that, compared to recent advancements in some $s$-MLLMs, including Imp, Moe-LLaVA, LLaVA-MoD, etc., LLaVA-KD still demonstrates superiority.
>
> **Q2.3 The performance of LLaVA-KD does not surpass that of LLaVA-MOD.**
>
> According to the test results reported by LLaVA-MoD in their paper (see Table 10 of the LLaVA-MoD paper), we conducted a comparison on the same benchmarks. We observed that when using teacher and student models of the same scale, our method demonstrates notable improvements. Specifically, **for the 1B and 2B student models, our average results across seven benchmarks indicate an advantage of 1.4 and 1.5, respectively.**
>
> Furthermore, we would like to emphasize that **the training data of LLaVA-MoD is nearly 3.8M more than our method (5M vs. 1.2M). Despite this, our method still demonstrates superior performance**, which is sufficient to prove our superiority.
>
> |           | LLM of Student Model | LLM of Teacher Model | GQA  | VizWiz | SQA  | TextVQA | MME  | MMB  | $\text{MMB}^{\text{CN}}$ | Avg   |
> |:---------:|:-------------:|:-------------:|:----:|:------:|:----:|:-------:|:----:|:----:|:------:|:-----:|
> | LLaVA-MoD | Qwen1.5-1.8B  | Qwen1.5-4B    | 58.7 | 34.6   | 67.9 | 57.7    | 67.6 | 64.9 | 60.7   | 58.9  |
> | LLaVA-KD  (Ours)  |          Qwen1.5-1.8B       |     Qwen1.5-4B          | 62.3 | 44.7   | 64.7 | 53.4    | 69.1 | 64   | 63.7   | **60.3**  |
> | LLaVA-MoD | Qwen1.5-0.5B  |      Qwen1.5-4B                     | 56.0   | 25.3   | 64.7 | 53.8    | 63.3 | 62.2 | 50.8   | 53.7  |
> | LLaVA-KD  (Ours)  |     Qwen1.5-0.5B           |       Qwen1.5-4B        | 59.6 | 35.9   | 60.6 | 49.9    | 64.5 | 60.1 | 55.5   | **55.2**  |

---

> ### Author Response · Authors · 2024-11-23
> **Response to Reviewer sLGE (2/2)**
>
> **Q2.4 The large MLLM used in this study has only 4B parameters, which may be insufficient to draw conclusions about scaling.**
>
>
> Thank you for your valuable comments. To address your concern, we attempted to use a 7B teacher model for distillation. As shown in the following table, we found that using the 7B teacher model for distillation resulted in an average performance drop of 0.5%. This observation aligns with the findings in [1], which claim: (1) "With increasing teacher size, ..., the trained student accuracy first increases and then decreases," and (2) "The teacher becomes so complex that the student does not have sufficient capacity or mechanics to mimic its behavior despite receiving hints." Therefore, when there is a significant size disparity between the teacher and student models, it can potentially hinder the transfer of knowledge. We believe that selecting an appropriate teacher model is crucial, rather than merely pursuing larger model sizes.
>
>
> | Teacher    | Student      | VQAv2 | GQA  | VisWiz | SciQA | TextVQA | MME | MMB |  $\text{MMB}^{\text{CN}}$ | POPE | MMMU | Avg   |
> |------------|--------------|-------|------|--------|-------|---------|------------|-------------|------------|------|-----------|-------|
> | Qwen1.5-4B | Qwen1.5-0.5B | 77.0    | 59.6 | 35.9   | 60.6  | 49.9    | 64.5       | 60.1        | 55.5       | 85.9 | 30.2      | 57.9  |
> | Qwen1.5-7B | Qwen1.5-0.5B | 76.9  | 59.1 | 35.9   | 59.5  | 49.3    | 63.1       | 58.0          | 54.4       | 86.6 | 31.4      | 57.4  |
>
> \[1\] Mirzadeh, Seyed Iman, et al. "Improved knowledge distillation via teacher assistant." Proceedings of the AAAI conference on artificial intelligence. 2020.
>
> **Q2.5 It is also unclear whether distilling a 1B model to another 1B model would be effective within this framework.**
>
> Generally speaking, the core objective of knowledge distillation is to transfer the knowledge of a large model to a smaller model, in order to maintain model performance as much as possible while reducing computational resource requirements. Typically, this means using a model with a larger number of parameters (e.g., 7B parameters) to distill a model with a smaller number of parameters (e.g., 1B parameters). If the two models have the same number of parameters (both 1B), then the main advantages of distillation cannot be realized.
>
> Currently, most successful distillation cases[2][3] are achieved in the transition from large models to small models, whereas in cases with the same number of parameters, the effects of distillation are usually not significant.
>
> We would like to further understand what the aim to verify with the 1B to 1B distillation. If you could provide more background information or specific research motivations, we would greatly appreciate it and could conduct more targeted discussions and experiments.
>
> **Q2.6 The authors do not report results for small MLLMs (0.5B and 1B) trained with the standard LLaVA procedure.**
>
>  In fact, TinyLLaVA (Qwen1.5-0.5B) and TinyLLaVA (Qwen1.5-1.8B) in Table1 are small MLLMs trained using the standard LLaVA procedure, following the Pre-Training and supervised Fine-Tuning strategy. Specifically:
> - With 1B parameters (using Qwen1.5-0.5B as the LLM), our method brings an improvement of 3.7% (57.3 vs. 61.0).
> - With 2B parameters (using Qwen1.5-1.8B as the LLM), our method brings an improvement of 5.9% (59.3 vs. 65.2).
>
> These results indicate that our method can enhance performance across models of different scales.
>
> |             | LLM          | #Samples | Training Strategy | AVG  |
> |:-----------:|:------------:|:--------:|:-----------------:|:--------:|
> | Tinyllava   | Qwen1.5-1.8B | 1.2M     | PT+SFT            | 59.3     |
> | LLaVA-KD-2B (Ours)  |        Qwen1.5-1.8B      |   1.2M        | DPT+SFT+DFT       | 65.2     |
> | Tinyllava   | Qwen1.5-0.5B |       1.2M   | PT+SFT            | 57.3     |
> | LLaVA-KD-1B  (Ours) |         Qwen1.5-0.5B     |      1.2M    | DPT+SFT+DFT       | 61.0     |
>
> \[1\] Tan, Shicheng, et al. "Gkd: A general knowledge distillation framework for large-scale pre-trained language model." arXiv preprint arXiv:2306.06629 (2023).
>
> \[2\] Gou, Jianping, et al. "Knowledge distillation: A survey." International Journal of Computer Vision 129.6 (2021): 1789-1819.
>
> \[3\] Zhao, Borui, et al. "Decoupled knowledge distillation." Proceedings of the IEEE/CVF Conference on computer vision and pattern recognition. 2022.

---

> ### Author Response · Authors · 2024-11-29
> **Reminder of the Discussion Period Deadline**
>
> Dear Reviewer sLGE
>
> We sincerely appreciate your time and effort. We have carefully considered your suggestions and have responded to each of your questions.
>
> As the Discussion phase is about to close, we would like to know if we have addressed your questions and concerns so far. We will continue to address your concerns until the Discussion phase deadline.
> If you could consider raising your evaluation of our paper after reviewing our responses, we would be very grateful.
>
> Thank you very much for your consideration.

---

> ### Author Response · Authors · 2024-12-01
> **Reminder of the Discussion Period Deadline**
>
> Dear Reviewer sLGE,
>
> Thank you for your time and valuable feedback. As the discussion phase is nearing its end, we remain open to addressing any remaining questions or concerns. We would greatly appreciate it if you could consider improving the evaluation after reviewing our responses. Thank you very much for your consideration.
>
> Sincerely, Paper #686 Authors

---

> ### Author Response · Authors · 2024-12-02
> **Reminder of the Discussion Period Deadline**
>
> Dear Reviewer sLGE,
>
> Thank you for your time and valuable feedback. As the discussion phase is nearing its end, we remain open to addressing any remaining questions or concerns. We would greatly appreciate it if you could consider improving the evaluation after reviewing our responses. Thank you very much for your consideration.
>
> Sincerely, Paper #686 Authors

---

### Official Review · Reviewer_xZ9J · 2024-11-05

**Soundness:** 2
**Presentation:** 3
**Contribution:** 2
**Rating:** 6
**Confidence:** 4

**Summary:**

The paper presents a knowledge distillation technique for multi-modal Large Language Models (MLLM) where the knowledge is transferred from a large MLLM (l-MLLM) to small MLLM (s-MLLM). Specifically, a three stage training is proposed (as opposed to two in conventional MLLM training) consisting of a) Distilled Pre-Training (DPT) to align visual-textual representations, b) Supervised Fine-Tuning (SFT) to equip the model with multimodal understanding, and c) Distilled Fine-Tuning (DFT) to further transfer l-MLLM capabilities. A Multimodal Distillation (MDist) and Relational Distillation (RDist) are used in both DPT and DFT stages to enhance the ability of s-MLLMs to process complex visual information. The proposed method is evaluated on a set of Benchmarks and VQA datasets to show improved performance w.r.to baseline and SOTA

**Strengths:**

1. The paper introduces a simple approach for knowledge distillation in MLLMs. While the components used such as KL divergence between teacher and student predictions [L_res in Eq. 2 and L_vis in Eq. 3, as in (Hinton, 2015), LLAVA-MoD are well-known], this method achieves superior results compared to current SOTA techniques employing knowledge distillation for MLLMs. Despite involving numerous hyperparameters (\alpha, \beta, \gamma, \alpha', \beta', \gamma'), the authors fixed these values across all experiments, demonstrating the method’s ability to generalize to new datasets without hyperparameter tuning.

2. The paper is well-written and easy to follow. The ablation study effectively demonstrates the significance of each component (MDist and RDist).

**Weaknesses:**

1. Techniques like Imp-2B (Shao et al., 2024) can achieve comparable results (<0.9) without relying on knowledge distillation. If models like Imp-2B can perform similarly without a pre-trained l-MLLM, what justifies the need for computationally intensive methods like the one proposed? The proposed approach would be more advantageous if it could achieve similar outcomes with significantly less data than Imp-2B, which currently requires only 300k more samples than the proposed method.

2. The authors claim that distilling visual tokens (Eqn. 3) is crucial for achieving strong results. However, as seen in Tables 3, 4, A3, and A4, the improvement from distilling visual tokens appears to be marginal (0.5%). I would appreciate further discussion or additional results to substantiate this claim, especially since it is one of the key contributions emphasized by the authors.

3. The proposed technique introduces an additional step (stage 3) compared to traditional MLLM training methods like LLAVA, potentially increasing the computational budget. While Table 5 provides a comparison, it is only against other KD techniques. I would prefer to see a comparison with all models of similar size (e.g., Imp-2B and others). Additionally, it is unclear why certain datasets, such as VQAv2, POPE, and MMMU, were excluded in Table 5.

After rebuttal: I appreciate the authors’ efforts in conducting additional experiments and find arguments 1 and 3 to be valid. Therefore, I am increasing my score to 6.

**Questions:**

I would like authors to address all 3 concerns outlined in the "Weaknesses" section

---

> ### Author Response · Authors · 2024-11-23
> **Response to Reviewer xZ9J (1/3)**
>
> We sincerely appreciate your constructive comments and suggestions. Thank you for your efforts in assessing and helping to improve the quality of our manuscripts. Below are our responses to your concerns:
>
> **Q1.1 If models like Imp-2B can perform similarly without a pre-trained l-MLLM, what justifies the need for computationally intensive methods like the one proposed?**
>
> Thank you for your comments. We would like to clarify three points:
> 1. In terms of performance comparison, Imp used 1M training data, which is 1.5 times the amount of LLaVA-KD training data. However, LLaVA-KD still demonstrated nearly a 1.0% performance advantage on 9 commonly used benchmarks. This indicates that LLaVA-KD exhibits stronger overall multimodal capabilities.
> 2. In terms of training data comparison: Imp-2B indeed performs well under the same scale of small models, but its performance is somewhat dependent on an additional 300k unavailable data.
> 3. In terms of computational resources, compared to the existing recent MLLMs methods based on knowledge distillation like LLaVA-MoD (16 A100(80G)), our method requires relatively lower computational resources (8 L40s(48G)).
>
> To prove our point, we trained the Imp-2B model based on the LLaVA1.5 dataset (the same as ours) according to the official code, as shown in the following table. We found that: (1) When using the LLaVA1.5-665k dataset for SFT, Imp-2B only showed an average performance of 63.7%, which is a 0.8% decrease compared to the original Imp-2B model. This indicates that the 300K training data contributed to the performance improvement of Imp-2B. However, this part of the data is currently not open-sourced, which is not conducive to further exploration by the community based on this method. In contrast, our method is based solely on the most common LLaVA training set, demonstrating greater flexibility in data dependency. (2) When trained on the same data, LLaVA-KD achieves notable performance, showing a 2.1% advantage.
>
> | Method      | LLM          | Training samples |                                                                                                                           | GPU           | Training Time (GPU Hours) | VQAv2 | GQA  | VisWiz | TextVQA | MME  | POPE | AVG   |
> |-------------|--------------|------------------|---------------------------------------------------------------------------------------------------------------------------|---------------|-----------------------------|-------|------|--------|---------|------------|------|-------|
> |             |              | PT               | FT                                                                                                                        |               |                             |       |      |        |         |            |      |       |
> | Imp-2B      | Qwen1.5-1.8B | LLaVA-1.5-558k   | 1M data: LLaVA-1.5-665k − 22K TextCaps + 32K OCR & chart data +30K captioning data + 300K conversation data (Unavailable) | 8 A100  (40G) | ~180                        | 79.2  | 61.9 | 39.6   | 54.5    | 65.2       | 86.7 | 64.5  |
> | Imp-2B      | Qwen1.5-1.8B | LLaVA-1.5-558k   | LLaVA-1.5-665k                                                                                                            | 8 A100  (40G)  | 140                         | 78.6  | 62.5 | 42.7   | 50.4    | 64.1       | 84.1 | 63.7  |
> | LLaVA-KD-2B (Ours) | Qwen1.5-1.8B | LLaVA-1.5-558k   | LLaVA-1.5-665k                                                                                                            | 8 L40s (48G)  | 320                         | 79.0    | 62.3 | 44.7   | 53.4    | 69.1       | 86.3 | 65.8  |
>
> In summary, our method not only outperforms Imp-2B in terms of performance but also does not rely on unavailable high-quality data and requires relatively lower computational resources. We believe that our method achieves a good balance in terms of performance, training data requirements, and computational resource requirements, providing a more practical solution for the development of s-MLLMs.

---

> ### Author Response · Authors · 2024-11-23
> **Response to Reviewer xZ9J (2/3)**
>
> **Q1.2 Further discussion on distilling visual tokens**
>
> In LLaVA-KD, distilling visual tokens encompasses two aspects. One is $L_{vis}$ in the MDist, where $s$-MLLM directly mimics the output of $l$-MLLM's visual tokens by constraining the KLD. The other is $L_{rel}$ in RDist, where $s$-MLLM simulates the relationship matrix of visual tokens from $l$-MLLM. As shown in the table below, we can observe that distilling visual tokens brings an average improvement of 0.7% during the DPT+SFT phase, and an additional 0.7% during the DFT phase, which means that visual token distillation results in a total performance improvement of 1.4% across the two distillation phases.
>
> Furthermore, the method of distilling visual tokens achieves performance improvement without significantly increasing computational resources. In contrast, other methods may require additional computational resources; for example, according to the Table1 in the Imp, the method adds 300k training data, resulting in a 1.1% performance improvement for Imp-3B.
>
> | Training stage | Distillation loss           | VQAv2 | GQA  | VisWiz | SciQA | TextVQA | MME  | MMB | $\text{MMB}^{\text{CN}}$ | POPE | MMMU | Avg   |
> |----------------|-----------------------------|-------|------|--------|-------|---------|------------|-------------|------------|------|-----------|-------|
> | DPT+SFT        | $L_{res}$                     | 73.8  | 57.8 | 25.6   | 62.8  | 47.1    | 59.7       | 55.9        | 49.3       | 85.5 | 31.6      | 54.9  |
> |        DPT+SFT          | $L_{res} + L_{vis} + L_{rel}$ | 74.6  | 57.8 | 28.8   | 61.2  | 49.1    | 59.9       | 56.9        | 51.6       | 84.3 | 31.4      | 55.6  |
> | DPT+SFT+DFT    | $L_{res}$                     | 76.8  | 59.6 | 36.4   | 59.1  | 50.2    | 64         | 57.6        | 52.7       | 85.8 | 30.1      | 57.2  |
> |          DPT+SFT+DFT      | $L_{res} + L_{vis} + L_{rel}$ | 77.0  | 59.6 | 35.9   | 60.6  | 49.9    | 64.5       | 60.1        | 55.5       | 85.9 | 30.2      | 57.9  |

---

> ### Author Response · Authors · 2024-11-23
> **Response to Reviewer xZ9J (3/3)**
>
> **Q1.3 A comparison with all models of similar size. Additionally, it is unclear why certain datasets, such as VQAv2, POPE, and MMMU, were excluded in Table 5.**
>
> In the original Table 5, we would like to make a comprehensive comparison with LLaVA-MoD, a MLLM method based on knowledge distillation (KD). LLaVA-MoD primarily reports its performance on seven benchmarks: GQA, VisWiz, SQA, VQA, MME, MMB, and $\text{MMB}^{\text{CN}}$. Therefore, we excluded VQAv2, POPE, and MMMU from Table 5 to maintain direct comparability with LLaVA-MoD.
>
> Based on your suggestion, we have added comparisons of LLaVA-KD with all similarly sized models, including Imp-2B and other relevant models, in the revised Table 5. Additionally, as shown in the table below, we have reported AVG (+POPE), AVG (+POPE+VQAv2), and AVG (+POPE+VQAv2+MMMU) to comprehensively evaluate the model's capabilities. It can be seen that our method demonstrates consistent superiority among models of the same scale.
>
>
> | Method         | #Params | #Samples | Training Time | AVG     | AVG (+POPE) | AVG (+POPE+VQAv2) | AVG (+POPE+VQAv2+MMMU)  |
> |:--------------:|:-------:|:--------:|:-------------:|:-------:|:-----------:|:-----------------:|:-----------------------:|
> | Imp-2B         | ~2B     | 1.5M     | ~180          | 58.9    | 62.4        | 64.3              | -                       |
> | Bunny-2B       |    ~2B     | 2.6M     | /             | 56.3    | 60.0        | 61.8              | -                       |
> | Mini-Gemini-2B |    ~2B     | 2.7M     | /             | 57.1    | 60.7        | -                 | -                       |
> | Tinyllava      |     ~2B    | 1.2M     | 105           | 53.9    | 57.6        | 59.3              | 56.8                    |
> | MoE-LLAVA      |     ~2B    | 2.2M     | /             | 55.3    | 59.2        | 61.1              | -                       |
> | LLaVA-MoD      | ~2B        | 5M       | 960           | 59.9    | 63.3        | -                 | -                       |
> | LLaVA-KD (Ours)      |  ~2B       | 1.2M     | 320           | **60.3**    | **63.5**        | **65.2**              | **62.1**                    |
> | SPHINX-Tiny    | ~1B    | 15 M     | /             | 51.2    | 55.1        | 57.3              | -                       |
> | TinyLLaVA      |   ~1B      | 1.2 M    | 52            | 51.1    | 55.2        | 57.3              | 54.7                    |
> | LLaVA-MoD      |    ~1B     | 5 M      | /             | 54.1    | -           | -                 | -                       |
> | LLaVA-KD (Ours)        |    ~1B     | 1.2 M    | 210           | **55.2**    | **59.0**        | **61.0**                | **57.9**                    |

---

> ### Author Response · Authors · 2024-11-29
> **Reminder of the Discussion Period Deadline**
>
> Dear Reviewer xZ9J:
>
> We sincerely appreciate your time and effort. We have carefully considered your suggestions and have responded to each of your questions.
>
> As the Discussion phase is about to close, we would like to know if we have addressed your questions and concerns so far. We will continue to address your concerns until the Discussion phase deadline. If you could consider raising your evaluation of our paper after reviewing our responses, we would be very grateful.
>
> Thank you very much for your consideration.

---

> ### Author Response · Authors · 2024-12-01
> **Reminder of the Discussion Period Deadline**
>
> Dear Reviewer xZ9J,
>
> Thank you for your time and valuable feedback. As the discussion phase is nearing its end, we remain open to addressing any remaining questions or concerns. We would greatly appreciate it if you could consider improving the evaluation after reviewing our responses. Thank you very much for your consideration.
>
> Sincerely, Paper #686 Authors

---

> > ### Author Response · Authors · 2024-12-02
> > **Reminder of the Discussion Period Deadline**
> >
> > Dear Reviewer xZ9J,
> >
> > Thank you for your time and valuable feedback. As the discussion phase is nearing its end, we remain open to addressing any remaining questions or concerns. We would greatly appreciate it if you could consider improving the evaluation after reviewing our responses. Thank you very much for your consideration.
> >
> > Sincerely, Paper #686 Authors

---

### Author Response · Authors · 2024-11-23

We sincerely thank all reviewers for the time and effort they have invested in the review process. Their positive feedback greatly encouraged us: Reviewer xZ9J affirmed that we achieved better results compared to the existing SoTA and noted that our method can be generalized to new datasets without adjusting the hyperparameters.; Reviewer sLGE considered our discussion on the optimization of small-scale MLLMs as "important and timely", Reviewer LdWi described our method as "well-motivated", pointing out that "tresearch on distillation methods for MLLMs remains largely unexplored." Reviewer Qt7K affirmed the novelty of our proposed method and found our performance "is quite impressive".

Meanwhile, based on the valuable feedback and suggestions from the reviewers, we have further improved our work in the following aspects:
1. We have added discussions and comparisons with the recent knowledge distillation-based MLLMs method: LLaVADI.
2. We have compared our method with existing LLM/MLLM distillation methods under the same experimental settings.
3. We have supplemented the ablation analysis of the hyperparameters in the model.
4. We have provided a deeper analysis of the effectiveness of visual distillation.
5. We have made a clearer explanation of the motivation for the proposed relational distillation algorithm.
6. We have corrected the statement comparing the performance with BLIP in the original subscription, and chose the most recent small scale MLLMs for a fair comparison.

Comprehensive responses to each point are detailed in the following columns.

---

### Author Response · Authors · 2024-11-25
**Please let us know whether we address all the issues**

Dear reviewers,

Thank you for the comments on our paper.

We have submitted the response to your comments and also more results are shown in the revised version. Please let us know if you have additional questions so that we can address them during the discussion period. We hope that you can consider raising the score after we address all the issues.

Thank you

---

### Author Response · Authors · 2024-11-26
**Reminder of the Discussion Period Deadline**

Dear reviewers,

Thank you for your time and valuable feedback.  Your comments are of great significance to the improvement of our work.

Before the ICLR discussion phase deadline, we are willing to address any remaining concerns you may have. If you could consider raising your evaluation of our paper after reviewing our responses, we would be very grateful.

Thank you very much for your consideration.

---

### Meta-Review · Area_Chair_8Aef · 2024-12-20

**Metareview:**

This paper proposes a framework to distill multimodal large language models (MLLMs). It consists of a three stages of training training: a) Distilled Pre-Training (DPT) to align visual-textual representations, b) Supervised Fine-Tuning (SFT) to equip the model with multimodal understanding, and c) Distilled Fine-Tuning (DFT) to further transfer l-MLLM capabilities. This is different from previous MLLM distillation works which usually have two stages. In addition, it proposes to use a Multimodal Distillation (MDist) and Relational Distillation (RDist) in both DPT and DFT stages to enhance the ability of s-MLLMs to process complex visual information. The experiments have shown that the proposed method is better than other MLLM distillation works.

This paper receives mixed reviews: 6 (xZ9J), 3 (sLGE), 6 (LdWi), 6 (Qt7K). They appreciate the contributions including 1) the paper is well written and easy to understand; 2) the investigated problem is important; 3) better results than other MLLM distilled works. However, the common concerns are 1) the overall novelty is small; 2) missing further discussion/ablation/benchmarks; 3) more training computation with additional stages; 4) some misleading claims; 5) missing experiments using larger teacher; 6) missing fair comparison under the same setting. After the rebuttal, most of the concerns by the reviewers were resolved, except the novelty issue and no gain by using larger LLMs. The AC has the same concerns on these. Using a larger (7B) teacher model leading to worse performance indicates that this distillation is not that effective nor robust. In addition, this paper only shows experiments on one MLLM architecture (LLaVA) and one LLM (Qwen1.5), which does raise the concern on generalization. Also, as a distillation paper, a more important point is to bridge the gap to the teacher model, but this paper focuses too much on the comparison with other distillation works and doesn't even provide the clear lower/upper-bound of the model.

Overall, the AC thinks this paper does have some unresolved issues and can be improved further before publication. So the AC recommends reject. The authors should take the feedback to improve the manuscript and resubmit to the future conferences.

**Additional Comments On Reviewer Discussion:**

For xZ9J: the main concerns are comparison with Imp-2B, distilling visual tokens are marginal, more computational budget, and missing datasets. The rebuttal resolved these concerns and the reviewer increased score to 6.
sLGE raises quite a few concerns: relational distillation motivation is unclear; the claim of superior performance over BLIP is misleading; missing benchmarks; missing experiments using larger teacher; unclear lower/upper-bound. It seems the main left concern is on the experiments using larger teacher model, and the reviewer kept the original score.
For LdWi, the main concerns are small novelty and missing fair comparison under the same setting. The fair comparison is somehow resolved by the rebuttal but still concern on the novelty part. To appreciate the additional experiments, the reviewer increased score from 5 to 6.
No big concern from Qt7K.

---

### Decision · Program_Chairs · 2025-01-22

Reject